# Magnetopause ripples going against the flow form azimuthally stationary surface waves

M. O. Archer [1✉], M. D. Hartinger[2], F. Plaschke [3], D. J. Southwood[1] & L. Rastaetter[4]

Surface waves process the turbulent disturbances which drive dynamics in many space, astrophysical and laboratory plasma systems, with the outer boundary of Earth's magnetosphere, the magnetopause, providing an accessible environment to study them. Like waves on water, magnetopause surface waves are thought to travel in the direction of the driving solar wind, hence a paradigm in global magnetospheric dynamics of tailward propagation has been well-established. Here we show through multi-spacecraft observations, global simulations, and analytic theory that the lowest-frequency impulsively-excited magnetopause surface waves, with standing structure along the terrestrial magnetic field, propagate against the flow outside the boundary. Across a wide local time range (09–15h) the waves' Poynting flux exactly balances the flow's advective effect, leading to no net energy flux and thus stationary structure across the field also. Further down the equatorial flanks, however, advection dominates hence the waves travel downtail, seeding fluctuations at the resonant frequency which subsequently grow in amplitude via the Kelvin-Helmholtz instability and couple to magnetospheric body waves. This global response, contrary to the accepted paradigm, has implications on radiation belt, ionospheric, and auroral dynamics and potential applications to other dynamical systems.

[1] Space and Atmospheric Physics Group, Department of Physics, Imperial College London, London, UK. [2] Space Science Institute, Boulder, CO, USA. [3] Space Research Institute, Austrian Academy of Sciences, Graz, Austria. [4] NASA Goddard Space Flight Center, Greenbelt, MD, USA. ✉email: m.archer10@imperial.ac.uk

Sharp discontinuities separating regions with different physical parameters are a key feature of space, astrophysical and laboratory plasmas. Their dynamics about pressure-balanced equilibrium, magnetohydrodynamic (MHD) surface waves, act as an efficient mechanism of filtering, accumulating and guiding the turbulent disturbances omnipresent between/through their respective systems. Surface waves have been observed and modelled within tokamak experiments[1], plasma tori surrounding planets[2], the solar atmosphere (e.g. in coronal loops[3]), the heliopause[4], accretion disks[5] and astrophysical/relativistic jets[6] to name a few. This makes understanding surface waves of universal importance.

While many of these environments can only be remote sensed, planetary magnetospheres (particularly that of Earth) provide the opportunity for in situ measurements of surface wave processes. The motion of the outer boundary of a magnetosphere, the magnetopause, is of primary importance in dictating global magnetospheric dynamics since it controls the flow of mass, energy and momentum from the solar wind into the terrestrial system having direct and indirect space weather impacts on the radiation belts, auroral oval and ionosphere[7–9]. Surface waves on a planetary magnetopause, which occupy the lower ends of the so-called ultra-low-frequency range (ULF; fractions of milliHertz to a few Hertz), are excited either by pressure imbalances (typically on the dayside) or flow shears (on the flanks)[10,11]. Magnetopause surface waves are thus similar to surface waves on bodies of water, which form due to and travel in the direction of the wind[12,13]. Since magnetopause surface waves impart momentum on the magnetosphere, the antisunward flow of the external driver—the shocked solar wind—has led to a well-accepted paradigm of the tailward propagation of outer magnetospheric ULF waves at all local times[14–17]. Surface waves may subsequently become non-linear via the Kelvin–Helmholtz instability at the magnetotail (when sufficient free energy is present to overcome magnetic tension or plasma compressibility) forming vortices that undergo magnetic reconnection, transporting mass across the boundary[18,19]. The paradigm of tailward propagation in magnetospheric dynamics is thought to hold even in response to the rather common impulsive events that drive intense space weather[20], for example, large-scale solar wind pressure pulses and shock waves[21,22] or smaller ($R_E$ scale or less) kinetically generated bow shock phenomena like magnetosheath jets[23]. The models predict an exception, in agreement with several observations, only at the early post-noon magnetopause, since pressure fronts aligned with the Parker spiral interplanetary magnetic field (IMF) strike this region before the pre-noon sector. Reported instances of sunward propagating ULF waves have been attributed to internal processes, such as energetic particle instabilities[24] or changes in the magnetotail configuration[25,26].

In physics, a common approach to understanding a dynamical system is to determine its normal modes. These form in a magnetosphere when system-scale MHD waves become trapped through reflection by boundaries or turning points. Transverse Alfvén waves, propagating along field lines due to magnetic tension forces, are reflected by the highly conductive ionosphere resulting in standing waves akin to those on a guitar string[27]. Fast magnetosonic waves, driven by correlated thermal and magnetic pressure gradients, trapped radially form so-called cavity/waveguide modes[28,29], somewhat similar to the resonances of wind instruments. These magnetospheric normal modes due to MHD body waves have been well-studied and tend to conform to the aforementioned paradigm[14–17]. However, it had long been proposed that magnetopause surface waves propagating along the terrestrial magnetic field in response to impulsive pressure variations might too reflect at the northern and southern ionospheres, forming a magnetopause surface eigenmode (MSE)

somewhat analogous to the vibrations of a drum's membrane[30]. The theory of these standing waves has been developed using ideal incompressible MHD in box model magnetospheres[31]. Despite their simplicity, these models have been able to reproduce many features captured by more advanced global MHD simulations[32]. For example, MSE frequencies near the subsolar magnetopause can be approximated in the limit $k_\phi \ll k_\parallel$ as (Eq. (6) of ref. [33], using pressure balance at the magnetopause)

$$\omega \approx k_\parallel \frac{B_{sph}}{\sqrt{\mu_0 \rho_{msh}}} \tag{1}$$

$$\approx k_\parallel \sqrt{2 \frac{\rho_{sw}}{\rho_{msh}}} v_{sw} \tag{2}$$

for angular frequency $\omega$, wavenumber $k$, magnetic field strength $B$, mass density $\rho$ and speed $v$ with subscripts $sw$, $msh$ and $sph$ corresponding to the solar wind, magnetosheath and magnetosphere, respectively. MSE thus constitute the lowest-frequency normal mode of the magnetospheric system, given the smaller phase speeds and wavenumbers than other modes. Indeed, Eq. (2) yields fundamental frequencies below 2 mHz and thus evanescent scales that highly penetrate the dayside magnetosphere[32,33]. However, MSE is thought to be strongly damped due to the finite thickness of the boundary, perhaps persisting for only a few wave periods[30,32,34]. Direct evidence of MSE was discovered only recently[35], likely due to the observational challenges in unambiguously demonstrating such a low-frequency normal mode has been excited. Fortuitous multi-spacecraft observations of the response to an isolated, broadband magnetosheath jet revealed narrowband magnetopause oscillations and magnetospheric ULF waves that were in excellent agreement with the theoretical predictions of MSE and could not be explained by other known mechanisms. These observations in the mid–late morning sector strikingly showed no azimuthal motion of the boundary despite the expectation that surface waves be advected tailward[10,20], hinting that this eigenmode may challenge the usual paradigm. It is currently unclear how to reconcile this with current models, especially since (unlike meridionally) there is no boundary azimuthally for surface waves to reflect against to reverse course.

In this paper, we address this conundrum by considering surface waves' energy flux through spacecraft observations, global MHD simulations, and analytic MHD theory in order to explain the resonant response of the magnetospheric system globally. We show that magnetopause surface waves propagate against the flow forming an azimuthally stationary wave across a wide local time range.

## Results

**Spacecraft observations.** We use Time History of Events and Macroscale Interactions during Substorms (THEMIS)[36] spacecraft observations from satellites A–E (THA–THE) during the previously reported event of MSE triggered by a magnetosheath jet on August 7, 2007. See the spacecraft observations section of 'Methods' for further details of instruments and techniques employed. The spacecraft were located at ~09:30 MLT (magnetic local time) in a string-of-pearls formation. For context, at 22:25 UT an isolated ~100 s magnetosheath jet occurred upstream of the magnetopause which was followed by a period of ~18 min with little pressure variations (demarked by vertical dotted lines) until another jet occurred at 22:45 UT (Fig. 2d of ref. [35]). The magnetopause moved in response to the jet, undergoing two boundary oscillations at 1.8 mHz corresponding to the fundamental mode MSE (Figs. 2g and 3b of ref. [35]). Figure 1 shows magnetospheric observations by the THA (panels a–i) and THE (panels j–r) spacecraft of the magnetic (panels a, j), velocity

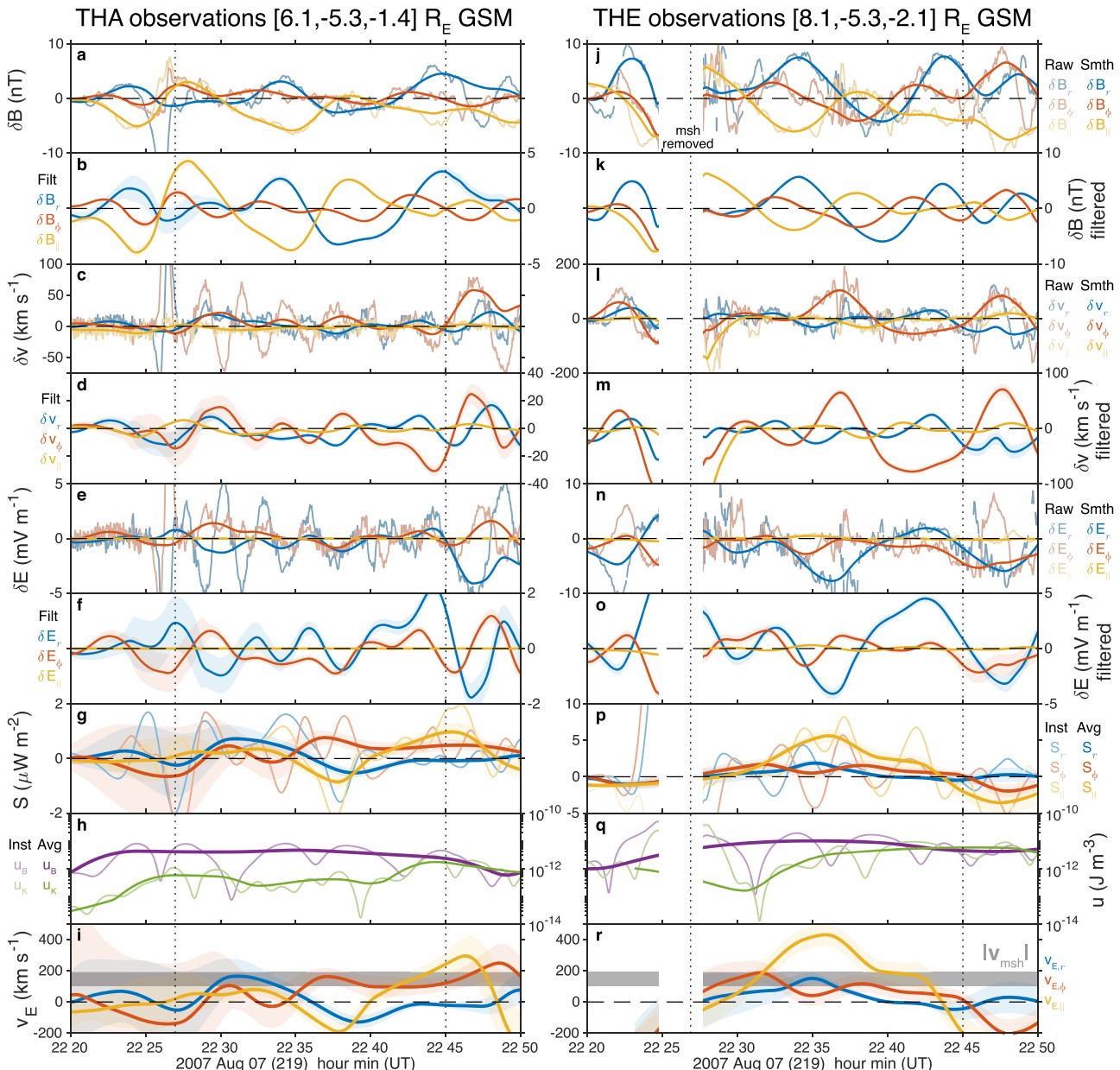

**Fig. 1 THEMIS spacecraft time-series observations in the magnetosphere.** Shown for THA (**a–i**) and THE (**j–r**). From top to bottom the first set of panels show perturbations in the magnetic (**a**, **b**, **j**, **k**), ion velocity (**c**, **d**, **l**, **m**) and electric (**e**, **f**, **n**, **o**) fields. In these vertical pairs, top panels show the raw (thin) and LOESS smoothed (thick) data, whereas the bottom panels show the latter once detrended. Subsequent panels depict the Poynting vector (**g**, **p**) and energy density (**h**, **q**), showing instantaneous (thin) and time-averaged (thick) values. Finally, the energy velocity (**i**, **r**) is shown compared to the absolute magnetosheath flow speed at THB (grey). Throughout, standard errors are indicated by shaded areas. Vertical dotted lines demark the times of little upstream pressure variations following the isolated impulsive jet that triggered this event.

(panels c, l) and electric fields (panels e, n). Dynamic spectra of these using the continuous wavelet transform can also be found in Supplementary Fig. 1 revealing the 1.8 mHz fundamental mode MSE (clearest in the compressional magnetic field components at both spacecraft) and 3.3 mHz second harmonic MSE (e.g., in the perpendicular components of the magnetic field), as well as a 6.7 mHz fundamental toroidal standing Alfvén wave at THA (azimuthal velocity/radial electric field)[35]. THD spacecraft observations proved similar to THE, and the other spacecraft encountered the magnetosheath too often for use here. We aim to measure the Poynting vector and corresponding energy velocity associated with MSE, concepts which are further explained in

Poynting's theorem for MHD waves section of 'Methods'. This necessitates extracting the associated wave perturbations from the data, removing noise and other signals as described in the time-based filtering section of 'Methods', resulting in the filtered magnetic (panels b, k), velocity (panels d, m) and electric fields (panels f, o) shown in Fig. 1. These are then used to determine energy densities and fluxes. Between the two dotted lines (which indicate the times of little upstream pressure variations), all spacecraft observed time-averaged Poynting vectors with components consistently azimuthally eastward and a slight tendency towards radially outwards too (panels g, p). This was also evident at the MSE frequencies in the Poynting vectors computed using

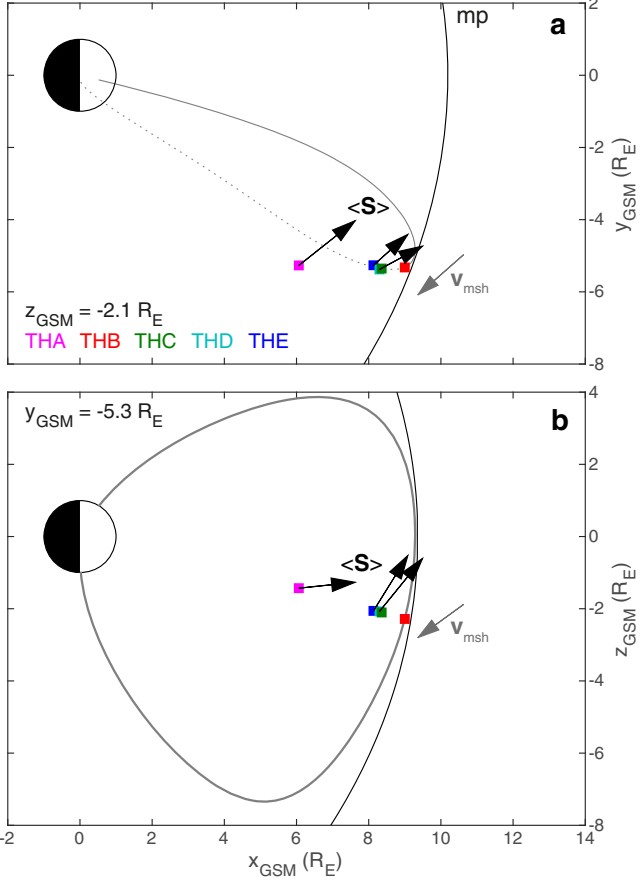

**Fig. 2 Directions of the time-averaged Poynting vectors at each THEMIS spacecraft location.** These are displayed in the $z_{GSM} = -2.1\,R_E$ (**a**) and $y_{GSM} = -5.3\,R_E$ (**b**) planes. Coloured squares represent the THEMIS spacecraft positions. Black arrows emanating from these indicate normalised Poynting vectors. Grey arrows depict the magnetosheath flow velocity direction. A representative geomagnetic field line (grey) and model magnetopause location (black) are also shown.

the wavelet transforms (see Fourier and wavelet techniques of 'Methods' for details) which are shown in Supplementary Fig. 1 in time–frequency and in Supplementary Fig. 2 as a function of frequency by averaging over the interval. The average Poynting directions at each spacecraft location are shown in Fig. 2, showing excellent agreement across all spacecraft in the equatorial plane (panel a). These observations show MSE do not conform to the typical ULF wave paradigm of tailward propagation[9,20] (waveguide modes' Poynting fluxes are directed tailward or have no net azimuthal component[37–39]; Kelvin–Helmholtz generated surface waves travel tailward and radiate energy into the magnetosphere[40,41]). The wave energy density (Fig. 1h, q) is dominated by the magnetic component, though the kinetic energy becomes comparable later in the interval. The waves' azimuthal energy velocity (Fig. 1i, r) is comparable to the flow speed in the magnetosheath (absolute value in grey) but oppositely directed, as indicated in Fig. 2a, suggesting the two forms of opposing energy flux might balance one another out and result in no net energy flow, i.e. an azimuthally stationary wave. This potentially may be behind the lack of azimuthal propagation in the observed boundary motion during this interval[35] and may hold the key to how MSE are even possible away from the noon meridian.

Since surface waves are formulated as collective magnetosonic waves on both sides of the boundary, the component of the Poynting vector towards the magnetopause might be understood in terms of the magnetosonic dispersion relation (Eq. (7) of ref. [10])

$$k_r^2 = -k_\phi^2 - k_\parallel^2 + \frac{\omega^4}{\omega^2 v_A^2 + c_s^2 \left(\omega^2 - \left[\mathbf{k} \cdot \mathbf{v}_A\right]^2\right)}, \quad (3)$$

where $\mathbf{v}_A$ is the Alfvén velocity and $c_s$ the speed of sound. Under incompressibility, the last term is negligible resulting in a purely imaginary $k_r$ and thus evanescence over similar scales to the length of the geomagnetic field lines. We relax this assumption and use a complex frequency $\omega = \omega_{\Re e} - i\gamma$ with damping rate $\gamma > 0$, since surface waves on a boundary of finite thickness are thought to be damped[30]. For the magnetospheric side, the phase of the last term in Eq. (3) is negative (approximately twice that of $\omega$ as the plasma beta is small) and thus $k_r^2$ has a negative imaginary component. This implies, for a physically reasonable solution with zero amplitude at infinity, $k_r$ should have a small real component pointed towards the magnetopause. We estimate that a damping ratio $\gamma/\omega_{\Re e} = 0.15$ should result in radial phase velocity components of ~10 km s$^{-1}$ (and between 1 and 60 km s$^{-1}$ for $\gamma/\omega_{\Re e} = 0.02 - 1$[32,34]), i.e. considerably smaller than the Alfvén speed of ~1000 km s$^{-1}$ and consistent with the average observed radial energy velocities of 9–46 km s$^{-1}$. This sense of propagation is opposite to what is expected for a Kelvin–Helmholtz unstable boundary, where the sign of $\gamma$ is reversed (being a growth rate) and thus results in energy radiating into the magnetosphere. By conservation of energy flux across the boundary, the Poynting vector component towards the boundary would imply that damped magnetopause surface waves lose some of their energy to the magnetosheath. This energy pathway would be in addition to the theorised irreversible conversion of surface wave energy to the Alfvén continuum[34].

THE and THD both observed significant field-aligned energy flux also, seemingly less prominent at THA. One might naively expect no field-aligned energy flux for a standing surface wave. However, considering this is a dynamical mode involving surface waves reflecting and interfering along the field under asymmetric conditions and driving, a resultant flux in this direction may be expected. For example, reflection at the ionospheres is neither perfect nor is the absorption north–south symmetric[42]. This will yield a superposition of standing and propagating waves with a null point, shifted slightly from the standing wave's nodes/antinodes, either side of which some resultant wave energy propagates to the respective ionospheres where it is dissipated[43]. The field-aligned flux at THE and THD peaks approximately one MSE bounce time after the driving jet. The corresponding energy velocity is consistent with the surface wave phase speed (Eq. (1)). It, therefore, seems plausible that these signatures are due to both the dipole tilt (Fig. 2b) resulting in different reflectances in both hemispheres and the localised driver exciting multiple harmonics with different relative phases causing shifts in the interference pattern. To the first point, the dipole tilt for this event was 17°, hence different conductances in the northern and southern ionospheres would be expected. Further, THA's footpoint (66° geomagnetic latitudes) could also have a different conductance to that for THD and THE (71°, i.e. near/in the auroral oval)[44], which could result in a difference in the proportion of wave energy reflected back to the spacecrafts' respective locations. To the second point, the wavelet transform demonstrates differences in the field-aligned Poynting fluxes for the two harmonics, most clearly shown in Supplementary Fig. 2 where averaging over time has been applied. The time-averaged Poynting flux at the fundamental MSE frequency of 1.8 mHz has a component in the direction of the geomagnetic field at all spacecraft,

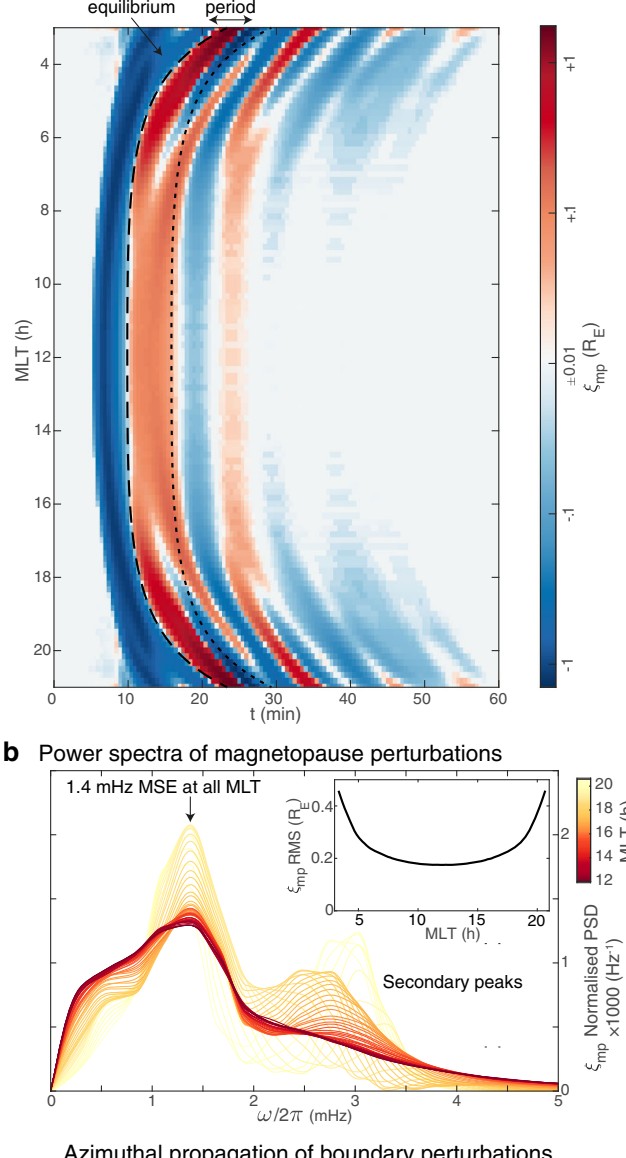

**a  Magnetopause normal displacement**

**b  Power spectra of magnetopause perturbations**

1.4 mHz MSE at all MLT

Secondary peaks

**Azimuthal propagation of boundary perturbations**

**c**  Raw  ±1σ
LOESS  ±1σ

**d**

**Fig. 3 Magnetopause motion in MHD simulation. a** Normal displacement with magnetic local time (MLT). **b** Spectra of the displacement normalised by total power for each MLT (colour scale), with the root mean squared (RMS) also given inset. Azimuthal slowness (**c**) and effective wavenumber (**d**) of filtered magnetopause perturbations for the raw data (lighter) as well as after applying MLT smoothing (darker), with corresponding standard errors shown in both cases as shaded areas.

indicating the spacecraft were all located above this harmonic's null point. The direction of the Poynting vectors agreed to within 26° and thus are consistent, taking noise into account. However, at the second harmonic MSE of 3.3 mHz, while the Poynting vectors' projections in the equatorial plane are similar (to within 9°), along the field we find that THE/THD observed southward fluxes, whereas at THA they were slightly northward (though not statistically significant from noise). A second harmonic wave has a node in displacement near the equator, thus at this frequency, the spacecraft observations are sensitive to which side of the null point they are located. In addition, smaller wavelength surface waves are less penetrating into the magnetosphere (Eq. (3)) which would weaken the signal at THA's location. We conclude that THA was very close to the second harmonic MSE's null point, whereas THE/THD were slightly below it. Nonetheless, the main result of interest in this paper, i.e. the fluxes radially and azimuthally, are in good agreement across all spacecraft at both frequencies. MSE's field-aligned energy flux may have implications on energy deposition in the ionosphere and warrants investigation in future work.

The above analysis was limited to a relatively short interval of confirmed MSE activity following an isolated magnetosheath jet. However, several other jets were also observed on this day and it was noted that similarly directed Poynting vectors followed many of them. We, therefore, take a wider interval and compute the time-averaged Poynting vector as a function of frequency, as detailed in the Fourier and wavelet techniques section of 'Methods'. This was performed for THA as it was the only spacecraft to experience uninterrupted magnetospheric observations. Supplementary Fig. 3 shows that at MSE frequencies (dotted lines) the radial and azimuthal Poynting vector components were statistically significant and in agreement with the previous results, namely outward and eastward. The parallel Poynting flux is positive at MSE frequencies indicating that THA was overall located above the respective null points of these waves[43]. We note that there is a reversal of the parallel Poynting flux around the local Alfvén mode frequency (6.7 mHz) thus unrelated to MSE. The typical tailward energy-flow paradigm emerges only at much higher frequencies (>10 mHz).

**Global MHD simulations**. To further test our hypotheses from the THEMIS observations, global MHD simulations of the global magnetospheric response to a 1 min large-scale solar wind density pulse are now employed (see Global MHD simulations in 'Methods'). This reproduces a previous simulation[32], where the subsolar response could only be explained by MSE and not other mechanisms. The normal displacement of the magnetopause in the XY plane is shown in Fig. 3a. This highlights the dayside magnetopause undergoes a strong compression when the pulse arrives, rebounds returning to equilibrium (dashed line) but overshoots, subsequently undergoing damped oscillations. Results are identical on both flanks due to the symmetry of the system and driver. The oscillations' primary frequency is 1.4 mHz at all local times (panel b), consistent with a fundamental MSE[32]. A secondary peak in the spectra, not previously reported, grows further downtail between 2.5 and 3.3 mHz. Both spectral peaks are associated with the damped oscillations and not the broadband initial compression/rebound motions, as checked by a wavelet transform. The secondary mode is likely due to (and at the frequency which maximises) the Kelvin–Helmholtz instability, given the increasing flow shear across the boundary down the flanks[45]. Both modes become larger in amplitude (panel b and inset) and persist longer (panel a) further downtail, though the primary mode is always dominant. This suggests that MSE at

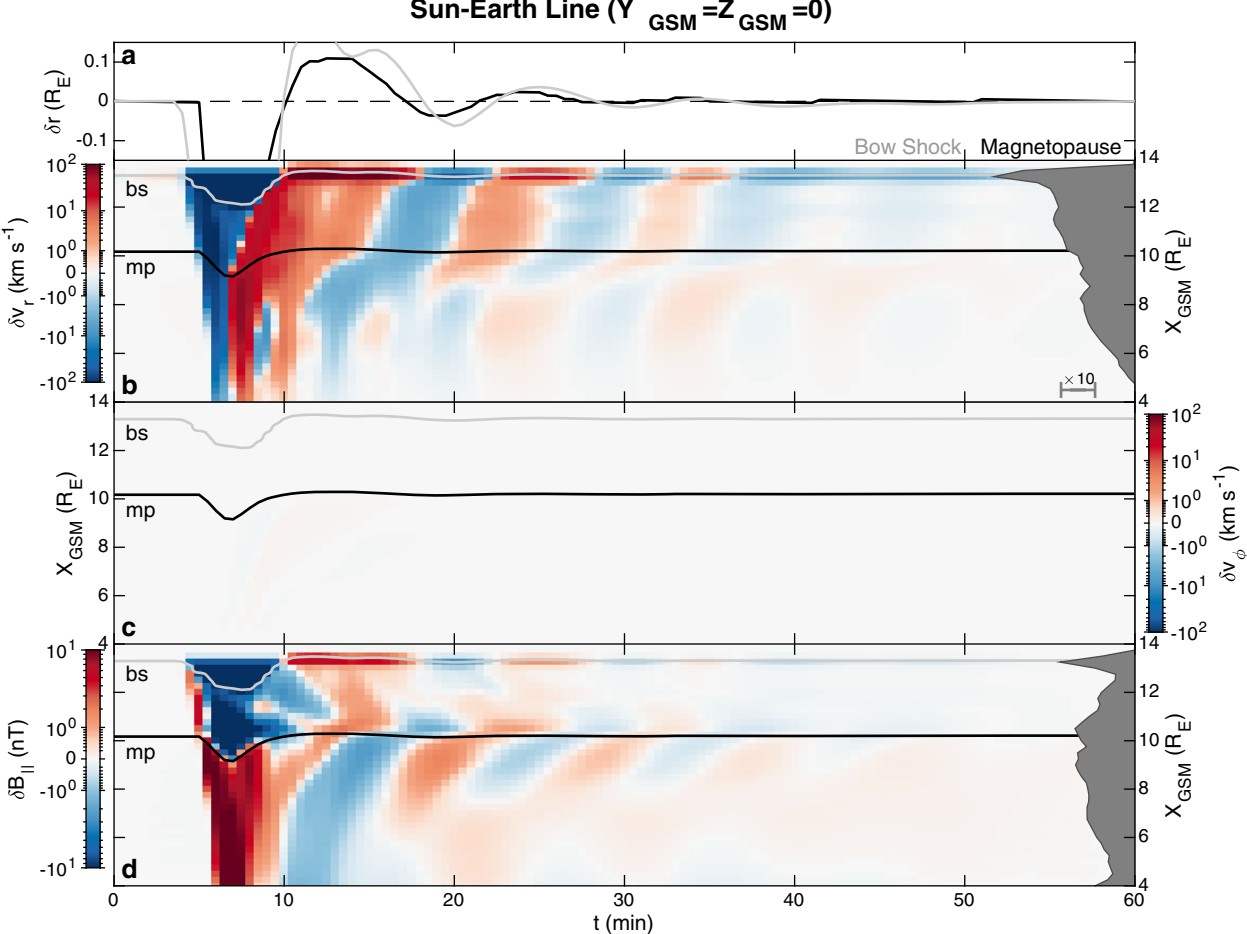

**Fig. 4 Unfiltered perturbations in MHD simulation along the Sun–Earth line.** Panel **a** shows motion of the bow shock (grey) and magnetopause (black) about equilibrium. Subsequent panels show perturbations in the **b** radial velocity, **c** azimuthal velocity and **d** compressional magnetic field components (note the bi-symmetric log scale). Median absolute perturbations by distance are displayed to the right as the dark grey areas on a logarithmic scale. The bow shock (light grey) and magnetopause (black) locations are also plotted.

1.4 mHz, which originates on the dayside magnetopause, seeds fluctuations that subsequently grow via Kelvin–Helmholtz in the flanks despite not being at the instability's peak growth frequency.

We now investigate the propagation of the 1.4 mHz magnetopause oscillations, extracted as described in the time-based filtering section of 'Methods'. From Fig. 3a it appears that across much of the dayside the waves do not propagate azimuthally (the phase fronts are vertical), whereas it is clear in the flanks that tailward propagating waves are present (inclined fronts, see also Supplementary Movie 1). Here, we quantify this propagation via the azimuthal slowness $s_\phi$ (reciprocal of apparent phase speed, see slowness in 'Methods') since the slowness vector is always normal to phase fronts[46]. The results are shown in Fig. 3c. This reveals that between ~09–15 h MLT the slowness is zero and thus the surface wave is apparently azimuthally stationary in this region. Further down both flanks though, the usual tailward motion is recovered. It may be instructive to express effective local azimuthal wavenumbers $m_{eff} = s_\phi \omega r_{mp}(\phi)$, shown in panel d ($r_{mp}(\phi)$ is the magnetopause geocentric distance at each azimuth). Care must be taken in interpreting these since the magnetopause crosses L-shells and is not azimuthally symmetric, so the dependence cannot be expressed simply as $\exp(im\phi)$ everywhere. Instead, a superposition of wavenumbers will be present, with $m_{eff}$ capturing the local azimuthal propagation of

the overall phase[47]. $|m_{eff}|$ is zero in the stationary wave region, rises slowly to ~0.5 by the terminator, then more rapidly increases to ~1 within a further 2 h of LT. This global structure cannot be attributed to the driver, since the intersection of the pressure pulse with the magnetopause on arrival spans 08–16 h MLT, i.e. larger than the stationary region.

We now look at the grid point data within the simulation. Supplementary Movie 1 shows the compressional magnetic field perturbations in the $XY$ and $XZ$ planes. Figure 4 shows boundary (panel a), radial (b) and azimuthal (c) velocity, and compressional magnetic field (d) perturbations along the Sun–Earth line These demonstrate that the arrival of the pressure pulse and inward magnetopause motion launches a compressional wave into the dayside magnetosphere which reflects at/near the simulation's inner boundary and subsequently leaks into the magnetosheath where it dissipates. This all happens within ∼2 min (i.e. before the magnetopause has finished rebounding) in agreement with the magnetosonic speed profile. Such a short timescale provides further evidence (in addition to that in ref. [32]) that the subsequent magnetopause oscillations on the dayside cannot be attributed to cavity/waveguide modes as the lowest-frequency (quarter wavelength[48]) mode should be ≳4 mHz. Azimuthal velocities are negligible, hence there is no evidence of toroidal Alfvén waves in this region. The MSE signatures instead are radial plasma motions and associated compressions/rarefactions of the

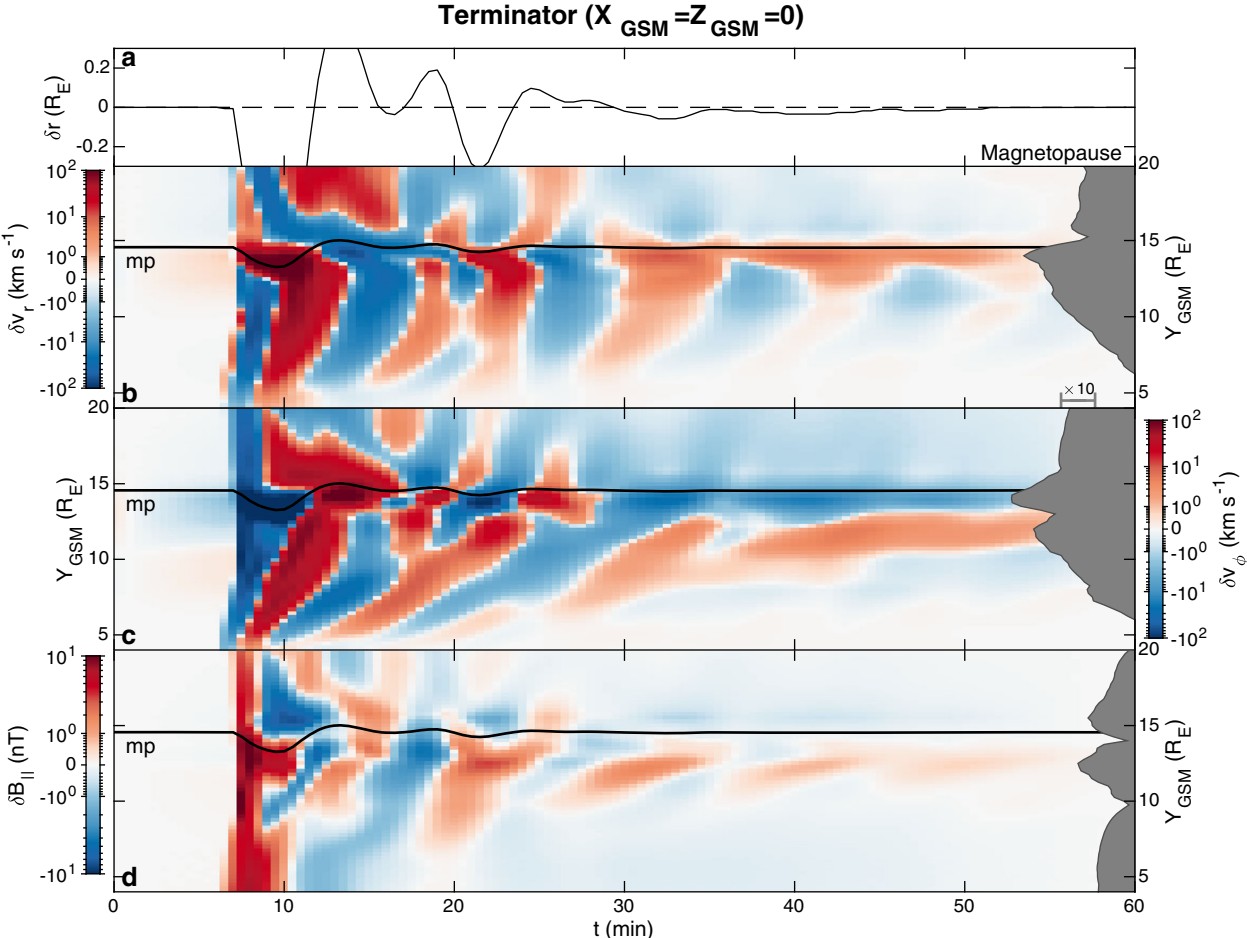

**Fig. 5 Unfiltered perturbations in MHD simulation along the equatorial terminator.** Formatting is the same as Fig. 4. **a** shows the motion of the magnetopause (black) about equilibrium. Subsequent panels show perturbations in the **b** radial velocity, **c** azimuthal velocity and **d** compressional magnetic field components (note the bi-symmetric log scale). Median absolute perturbations by distance are displayed to the right as the dark grey areas on a logarithmic scale. The magnetopause (black) location is also plotted.

magnetic field, both of which decay in amplitude with distance from the magnetopause as indicated by the median absolute perturbations (grey areas in Fig. 4). The phase fronts, however, are not purely evanescent and can be seen propagating towards the magnetopause on the magnetospheric side. This occurs rather slowly though at around 30–40 km s$^{-1}$ near the boundary, in agreement with the estimates due to damping made earlier. Deeper into the magnetosphere more evanescent and less propagating behaviour is found, as expected from Eq. (3) due to the greater Alfvén speeds. The magnetic field perturbations on either side of the boundary are in approximate antiphase with one another throughout the dayside (note the magnetopause thickness in the simulation is ~1.5 R$_E$, considerably larger than in reality since gyroradius-scales are not resolved[49]). There is evidence of large-scale bow shock motion related to MSE, a consequence that has not been considered before. At the subsolar point (see Fig. 4) the bow shock lags the magnetopause motion by ~1 min, consistent with the fast magnetosonic travel time through the magnetosheath, confirming that the resonance is occurring at the magnetopause and subsequently driving the shock oscillations. This lag occurs because magnetosheath plasma is highly compressible[31,33], thereby deviating from the evanescent behaviour expected under incompressibility. Since both the magnetopause and bow shock move asynchronously, the patterns present in the subsolar magnetosheath are somewhat complicated. These could be explored further in the future.

In Supplementary Movie 1, magnetic field perturbations in the equatorial plane are in phase across much of the dayside showing little evidence of azimuthal propagation, confirming that $k_\phi$ is much smaller than $k_r$ and $k_\parallel$. Tailward travelling disturbances can be seen emanating from the oscillations near the dayside magnetopause only at ~09 h and 15 h MLT, hence are associated with the propagating surface waves discussed earlier. Supplementary Movie 2 separates out these two regimes for further clarity. Further down the flanks, at ~07 h and 17 h MLT, structure normal to the magnetopause emerges with strong peaks/troughs ~2R$_E$ inwards from the boundary. Figure 5 shows cuts along the terminator. This reveals, in addition to the surface waves, the presence of a quarter wavelength waveguide mode[39,48] (at the magnetopause there is a $\delta v_r$ antinode and $\delta B_\parallel$ node; $\delta B_\parallel$ exhibits nodal structure radially). The waveguide mode couples to a toroidal Alfvén mode[50] at $Y_{GSM}$ ~11.5 R$_E$ ($\delta v_\phi$ antinode). These two modes occur at the same ~ 10 min period as the surface waves that originate at the subsolar point. Therefore, MSE can couple to body eigenmodes in regions of the magnetosphere where their frequencies sufficiently match (checked through time-of-flight estimates). Magnetospheric Alfvén speed profiles are highly variable and significantly alter the eigenfrequencies of both body modes[51], thus we expect that whether and where this coupling may occur will vary substantially.

In the *XZ* plane, Supplementary Movie 1 reveals that the ~ 10 min period oscillations do not extend beyond the cusps

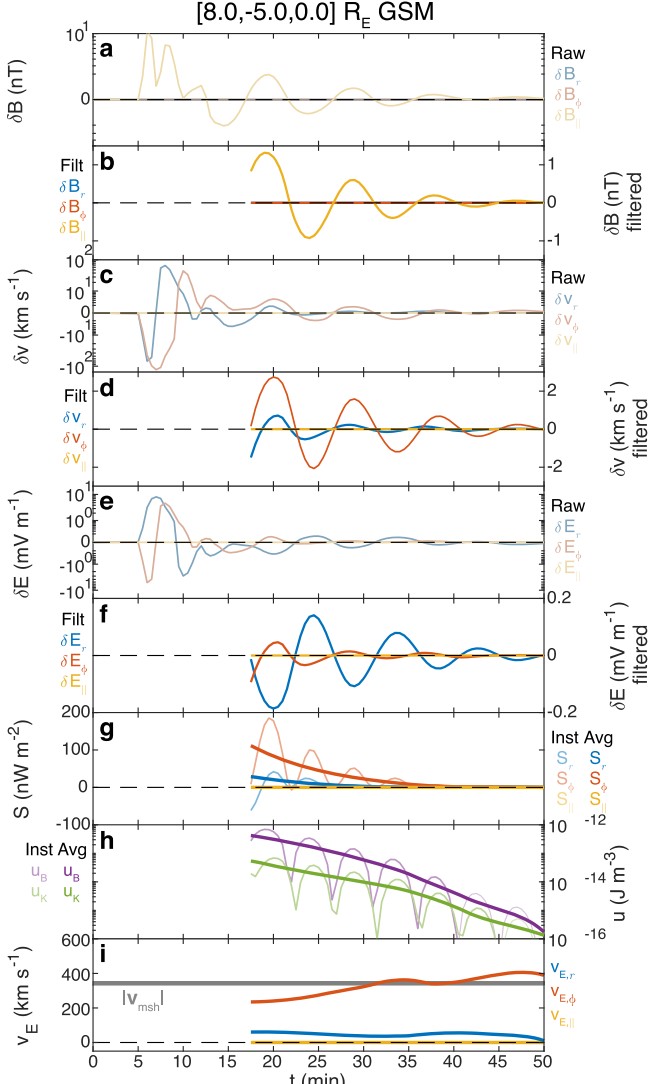

**Fig. 6 Virtual spacecraft observations within MHD simulation.** Displayed in a similar format to Fig. 1. From top to bottom the first set of panels show perturbations in the magnetic (**a**, **b**), ion velocity (**c**, **d**) and electric (**e**, **f**) fields. In these vertical pairs, top panels show the raw data, whereas the bottom panels show the filtered data. Subsequent panels depict the Poynting vector (**g**) and energy density (**h**), showing instantaneous (thin) and time-averaged (thick) values. Finally, the energy velocity (**i**) is shown compared to the absolute magnetosheath flow speed (grey). Note the bi-symmetric log scale on panels **a**, **c**, **e**.

into the northern and southern tail lobes. The waves are thus confined to closed magnetic field lines, further backing the surface eigenmode interpretation.

We finally study energy propagation throughout the simulation. Figure 6 shows results from a virtual spacecraft in the magnetosphere close to the boundary at roughly the same location as THE. Computing the Poynting vector as before (panel g) shows it to be directed azimuthally towards the subsolar point and slightly radially outwards, similarly to the observations. This corresponds to an energy velocity (panel i) approximately equal but opposite to the background magnetosheath flow speeds (grey) at this local time, like in the observations.

Figure 7 shows equatorial maps of the time-averaged Poynting (panel a) and advective (panel b) energy fluxes as well as the sum of the two (panel c). Within the magnetosphere, the Poynting vectors are directed azimuthally towards the subsolar point across the

entire dayside, flipping direction at around the terminator to recover the more usual tailward energy flux associated with Kelvin–Helmholtz generated surface waves and waveguide modes[37,40,41]. Later within the simulation, however, this point of reversal does slowly move slightly towards noon by ~1 h of MLT on both flanks as the wave energy dissipates. A component directed towards the magnetopause is also present across the dayside until well into the flanks and the continuity of this energy flux into the magnetosheath is apparent. The advective energy flux in Fig. 7b consists predominantly of the tailward flow throughout the magnetosheath. Therefore, the sum of the two clearly shows across the dayside that energy fluxes tangential to the magnetopause are in opposition to one another on either side. A small amount of energy flows across the boundary from the magnetosphere into the magnetosheath, which will then be swept downtail.

We, therefore, investigate the potential balance of tangential energy fluxes on either side of the magnetopause in Fig. 8. At each local time, we construct rays normal to the equilibrium magnetopause and interpolate the time-averaged Poynting (panel a) and advective (panel b) energy fluxes, taking the component tangential to the boundary. Integrating these along the normal, we arrive at panel c showing the total tangential energy flux across both sides of the magnetopause. This demonstrates the tailward energy flow due to advection (green) and opposing Poynting flux (purple) across the dayside. Taking the sum of these shows that they cancel out between 08:40 and 15:20 MLT, i.e. the local time range for which the magnetopause oscillations were found to be azimuthally stationary. This range is stable in time for the duration of the oscillations. The results, therefore, demonstrate that the stationary nature of MSE azimuthally is due to a balance of the surface wave Poynting flux directed towards the subsolar point opposing the tailward magnetosheath flow. Outside of this region, however, even when the Poynting flux is in opposition to the magnetosheath flow it is unable to overcome advection and thus travelling surface waves result.

**Analytic MHD theory.** Finally, we look to incompressible MHD theory (where $k_r^2 + k_\phi^2 + k_\parallel^2 = 0$ for surface waves[10,31]) to understand this picture of the energy flow and azimuthal propagation present within MSE. We consider a fundamental mode magnetopause surface wave of amplitude $A$ in displacement within a box model magnetosphere with homogeneous half-spaces as depicted in Fig. 9a. The azimuthal component of the Poynting vector at the equator on the magnetosphere side of the boundary for northward IMF is given by (Eq. (15) of ref. [40])

$$\left\langle S_{\phi,sph} \right\rangle = A^2 \omega k_\phi \frac{k_\parallel^2}{k_\phi^2 + k_\parallel^2} \frac{B_{0,sph}^2}{2\mu_0} \exp\left(-2\left|\Im m(k_r)\right|\left|r - r_{mp}\right|\right)$$

(4)

with its equivalent on the magnetosheath side being $B_{0,msh}^2/B_{0,sph}^2$ times this and thus negligible. The wave energy densities are (following Eqs. (11) and (13) of ref. [40])

$$\left\langle u_{sph} \right\rangle \approx \left\langle u_{B,sph} \right\rangle = \frac{B_{0,sph}^2}{4\mu_0} A^2 \frac{k_\parallel^4}{k_\phi^2 + k_\parallel^2} \exp\left(-2\left|\Im m(k_r)\right|\left|r - r_{mp}\right|\right)$$

$$\left\langle u_{msh} \right\rangle \approx \left\langle u_{K,msh} \right\rangle = \frac{1}{4} \rho_{0,msh} \omega^2 A^2 \frac{2k_\phi^2 + k_\parallel^2}{k_\phi^2 + k_\parallel^2} \exp\left(-2\left|\Im m(k_r)\right|\left|r - r_{mp}\right|\right)$$

(5)

Constructing the net energy velocity of the surface wave and

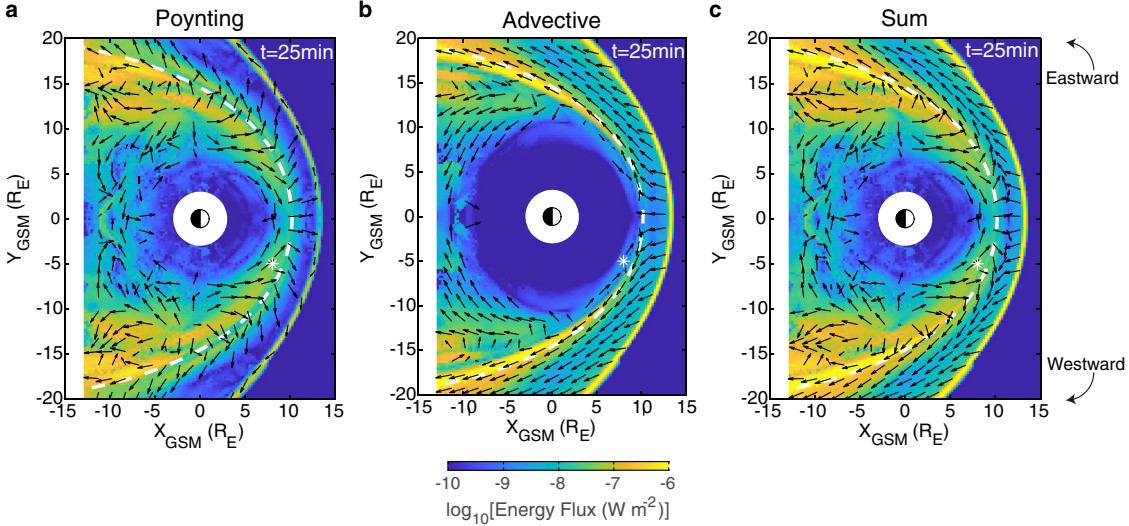

**Fig. 7 Wave energy flux maps.** Panels show time-averaged wave Poynting (**a**), advective (**b**) and total (**c**) energy fluxes. Magnitude (colour) and direction (arrows) are shown along with the equilibrium magnetopause (white dashed) and virtual spacecraft location from Fig. 6 (star).

simplifying using Eq. (1) gives

$$\mathbf{v}_{E,tot} = \frac{\langle \mathbf{S}_{sph} \rangle + \langle \mathbf{S}_{msh} \rangle + \langle u_{msh} \rangle \mathbf{v}_{0,msh}}{\langle u_{sph} + u_{msh} \rangle} \quad (6)$$

$$\approx \frac{\omega k_\phi k_\parallel^2 \frac{B_{0,sph}^2}{2\mu_0} + \frac{1}{4}\rho_{0,msh}\frac{B_{0,sph}^2}{\mu_0\rho_{0,msh}}k_\parallel^2\left(2k_\phi^2 + k_\parallel^2\right)v_{0,msh}}{\frac{B_{0,sph}^2}{4\mu_0}\left[k_\parallel^4 + \omega^2\frac{k_\parallel^2}{\omega^2}\left(2k_\phi^2 + k_\parallel^2\right)\right]}\hat{\phi} \quad (7)$$

$$\approx \left[\frac{\omega k_\phi}{k_\phi^2 + k_\parallel^2} + \frac{2k_\phi^2 + k_\parallel^2}{2\left(k_\phi^2 + k_\parallel^2\right)}v_{0,msh}\right]\hat{\phi} \quad (8)$$

By setting this to zero, i.e. no net azimuthal energy flow, and solving for real azimuthal wavenumbers yields the requirement

$$\frac{\omega}{k_\parallel} \geq \sqrt{2}v_{0,msh} \quad (9)$$

This sets a limit on where it is possible for surface wave energy to be trapped locally due to the speed of the adjacent magnetosheath. We can frame this limit purely in terms of solar wind and magnetosheath conditions using Eq. (2) as

$$v_{0,msh} \leq \sqrt{\frac{\rho_{sw}}{\rho_{msh}}}v_{sw} \quad (10)$$

According to gas-dynamic models of magnetosheath plasma conditions[52], this is satisfied for 08:40–15:20 h MLT, in excellent agreement with the stationary region in the global MHD simulation. This extent should vary only slightly with solar wind conditions (based on previous magnetosheath and MSE variability studies[33,53]), however, future work could test this.

## Discussion

In this paper, we show that the recently discovered magnetopause surface eigenmode (MSE), the lowest-frequency normal mode of a magnetosphere, does not conform to the well-established paradigm in global magnetospheric dynamics of tailward propagation. Multi-spacecraft observations, global MHD simulations and analytic MHD theory are employed. Both the observations and simulation show Poynting vectors in the magnetosphere which point towards the subsolar point across the dayside,

contrary to current models of the magnetospheric response to impulsive driving[20]. This energy flux thus opposes advection by the magnetosheath and we find from the simulation that these two cancel one another in the region 09–15 h magnetic local time, resulting in an azimuthally stationary surface wave. Outside of this region, however, the waves travel tailward. Considering surface wave energy fluxes in a simple box model of the magnetosphere shows excellent agreement with the simulation on the conditions required for a stationary wave to be possible. Our conclusions are summarised in Fig. 9 within this box model. When an impulsive solar wind transient arrives at the magnetopause, its broadband nature excites surface waves on the boundary with a wide range of frequencies $\omega$ and wavevectors $\mathbf{k}$. The boundary conditions at the northern and southern ionospheres quantise the possible values of $k_\parallel$ largely determining $\omega$, however, $k_\phi$ will be unconstrained as depicted in panel a. For large magnetosheath flow speeds (panel b), none of the excited wavevectors are able to compete with advection and the resultant motion is tailward, in line with expectations. In the regime of small magnetosheath flow speeds (panel c), however, there exists an excited $k_\phi$ in opposition to the magnetosheath flow which is able to exactly balance its advective effect. This leads to surface wave energy being trapped locally as an azimuthally stationary wave. All waves of other $k_\phi$ will be lost down the tail. This picture not only explains the global propagation of magnetopause surface waves but also how MSE on the dayside can seed fluctuations into the magnetospheric flanks. The simulation shows that these seeded waves which originate on the dayside subsequently grow in amplitude via the Kelvin–Helmholtz instability, despite being at a lower frequency to its intrinsic one, and may couple to cavity/waveguide and Alfvén modes in regions of the magnetosphere where their frequencies match. This reveals MSE's effects are not confined merely to the dayside (standing) region, instead having global effects on the magnetosphere as its most fundamental normal mode.

The cartoon highlights that, at each location on the boundary, after the other (blue) wavevectors have been swept downtail and the boundary has formed its resonance, the physics of the azimuthally stationary surface wave is confined to a small local time region, i.e. a single meridian of geomagnetic field lines. While the initial perturbation on the boundary and the corresponding transient response will depend on the specifics of the driving

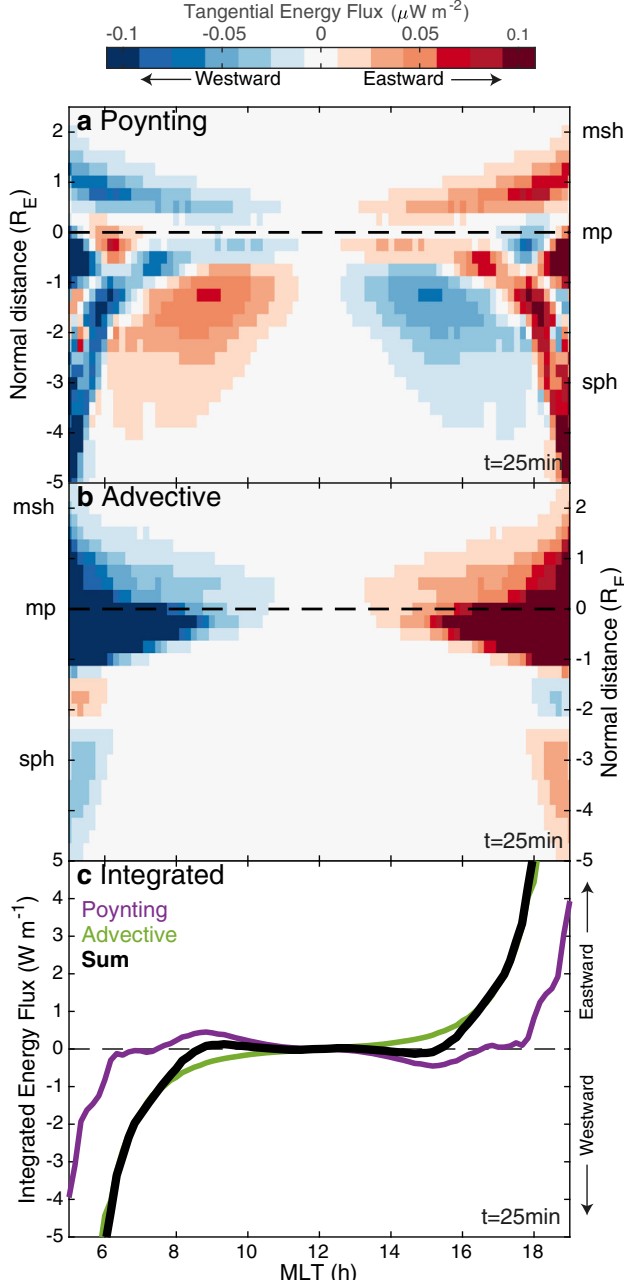

**Fig. 8 Wave energy fluxes tangential to the magnetopause.** Panels show Poynting (**a**) and advective (**b**) energy fluxes tangential to the magnetopause along magnetopause normals. Integrals along the normal are shown in panel **c** for the Poynting (purple) and advective (green) fluxes along with their sum (black).

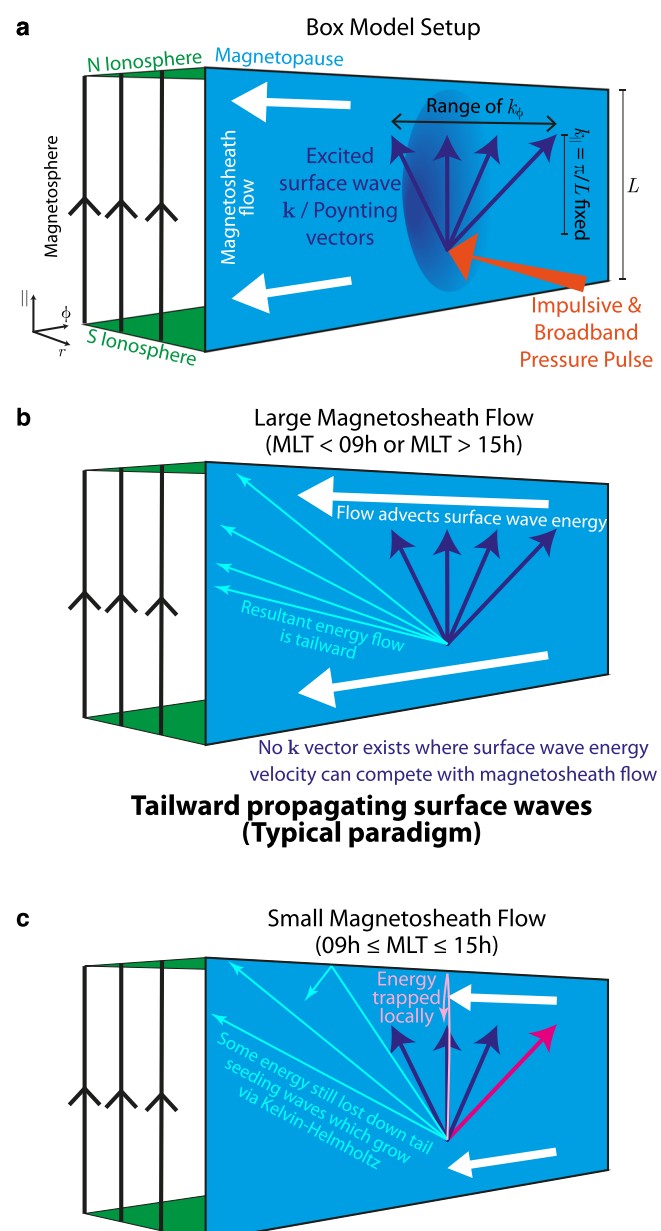

**Fig. 9 Cartoon illustrating the results of this study. a** shows the box model magnetosphere, magnetosheath flow (white), and surface mode wavevectors (dark blue) excited by the pressure pulse (orange). Subsequent panels depict the resultant energy flow (lighter coloured arrows) of the surface mode wavevectors for (**b**) large and (**c**) small magnetosheath flows.

pressure variation (scale sizes, location of impact etc.), one can simply decompose the initial perturbation at each local time into the normal modes along the field (MSE) and this should entirely dictate the subsequent resonant response at that local time. Indeed, the local boundary motion and Poynting fluxes were in agreement across both observations and simulations despite different scale size drivers being leveraged. The locality to the physics means that the azimuthally stationary wave should be limited to the local times in which the driver impacted the magnetopause, hence the scale of the driver in azimuth (within the 09–15 h local time range) would be imprinted in the stationary waves excited.

These results raise the question why only tailward propagating dynamics are reported in current models and observations of the magnetospheric response to impulsive driving[20]. It is clear that the models do not incorporate the possibility of surface wave reflection due to bounding by the ionosphere, which is key to our results, since while this was proposed long ago[30] it has only recently been discovered[2,35]. MSE constitute the lowest possible frequency normal mode of the magnetospheric system and its fundamental frequencies can often be fractions of milliHertz[33].

Such long-period narrowband waves are challenging to identify observationally in general, either by orbiting spacecraft or ground-based measurements, due to potential spatio-temporal mixing[54] and the difficultly in distinguishing from turbulence/noise[55,56]. For these reasons global magnetospheric dynamics and their associated ULF waves have often concentrated on the continuous pulsation (Pc) bands above 2 mHz[57], which would not typically incorporate the effects presented here. Furthermore, Fig. 9 shows that the majority of surface wave energy excited by the driver does still propagate tailward, with only a small amount being trapped locally that propagate against the flow. Therefore, if further upstream driving by pressure variations occurs during these oscillations, the superposition of waves present could easily mask the sunward Poynting vectors associated with MSE thereby showing only a net tailward energy flow. Future work is required in developing less restrictive observational criteria for the detection of MSE in general and to undertake statistical studies of MSE occurrence to better understand how common this mode, and the results presented on its energy flow, may be in reality to the variety of impulsive drivers that impact on geospace.

The global waves associated with this normal mode of Earth's magnetosphere, possible due to the surface wave propagation against the magnetosheath flow, will have important implications upon radiation belt dynamics[7,8]. The large-scale oscillations of the magnetopause may cause the further shadowing of radiation belt electrons than predicted simply by a pressure-balanced quasi-static response to the driver. Furthermore, MSE's ULF wave signatures present coherent and slowly varying perturbations in compressional magnetic fields and azimuthal electric fields which deeply penetrate across the entire dayside magnetosphere, which may be ideal for the drift-resonant interaction and/or radial diffusion of radiation belt particles. However, current methods of understanding these processes are suited only to the inner magnetosphere since they assume azimuthal symmetry, therefore, more work is required in assessing the impact on the radiation belts of this normal mode and asymmetric outer magnetospheric waves in general. While the observations confirm significant energy flow along the magnetic field towards the polar regions, the wide region of stationarity from the simulations suggests weak coupling of the surface waves to the Alfvén mode across the dayside. This implies MSE have auroral, ionospheric, and ground magnetometer signatures unlike other known ULF waves, with these remaining poorly understood[2,35]. In the flanks, however, it seems likely that MSE-seeded waves could at times easily be mistaken for intrinsic Kelvin–Helmholtz waves or waveguide modes, despite the origin of the fluctuations on the dayside as shown in the simulation. These factors may be why previous ground-based searches, through widely used diagnostics for other wave modes, called MSE's existence into question[58,59]. The work thus highlights that care needs to be taken in understanding the mechanisms which result in various dynamical modes in near-Earth space since they can all be intimately coupled.

Surface waves are known to be present at the other planetary magnetospheres[60,61], which span a vast range of sizes, morphologies and plasma conditions[62]. The surface eigenmode in principle should be a universal feature of boundaries in magnetospheres[30], and thus the simple analytic theory presented here (in the magnetospheric reference frame) may be instructive in assessing where and in what frequency ranges these fundamental dynamics of the boundary may be prevalent at other environments. The simple predictions could then be compared to tailored global MHD simulations of these systems as well as spacecraft observations.

Many other space and astrophysical systems too exhibit surface waves where, like in the case of a magnetopause, substantial background flows may be present. A notable example is the sausage and kink modes of coronal loops, which share many conceptual similarities to the surface eigenmodes—they are standing (though sometimes propagating) transverse oscillations of the dense flux tubes in coronal active regions, anchored on both ends by the chromosphere, excited by loop displacements from coronal eruptions or shear flows in coronal plasma non-uniformities[3,63]. Asymmetric and/or inhomogeneous flows around/along these structures affect surface wave evolution, with important space weather consequences such as causing coronal mass ejections to turn away from their original propagation direction[64], though these effects are not typically incorporated into models of coronal loop oscillations. Our results from in situ observations at the magnetopause (not possible for the corona and other space/astrophysical environments) challenge the paradigm that surface waves necessarily propagate in the direction of the driving flow/pressure, as when discontinuities are bounded the trapping of surface waves may occur in opposition to advective effects, allowing these waves to form across broader regions and to persist longer than would otherwise be expected. The work may therefore have insights into the structure and stability of these universal dynamical modes elsewhere.

## Methods

**Poynting's theorem for MHD waves.** Energy conservation for MHD wave perturbations (denoted by $\delta$'s with subscript 0's representing equilibrium values) involves the wave energy density

$$u = u_B + u_K = \frac{|\delta\mathbf{B}|^2}{2\mu_0} + \frac{1}{2}\rho_0|\delta\mathbf{v}|^2 \qquad (11)$$

(consisting of magnetic $u_B$ and kinetic $u_K$ contributions) and wave energy flux given by the Poynting vector[65]

$$\mathbf{S} = \frac{\delta\mathbf{E} \times \delta\mathbf{B}}{\mu_0} \qquad (12)$$

where $\mathbf{E}$ is the electric field. Time-averaging and taking their ratio yields the so-called energy velocity

$$\mathbf{v}_E = \frac{\langle\mathbf{S}\rangle}{\langle u \rangle}, \qquad (13)$$

equivalent to the group velocity for stable waves[66]. In a moving medium, wave energy advects with the background plasma, giving an additional flux $u\mathbf{v}_0$. These principles are applied throughout.

**Spacecraft observations.** Observations in this paper are taken from the Time History of Events and Macroscale Interactions during Substorms (THEMIS[36];) spacecraft taken during the previously reported interval of MSE[35]. The five spacecraft were close to the equilibrium magnetopause in a string-of-pearls formation. Data from the fluxgate magnetometer (FGM)[67], electrostatic analyser (ESA)[68] and electric field (EFI)[69] instruments are used. Note for the latter, we use the $\mathbf{E} \cdot \mathbf{B} = 0$ approximation (valid over ULF timescales) for THD and THE to replace the measured axial fields at each time, however, the instrument was not yet deployed by THA so $\mathbf{E} = -\mathbf{v} \times \mathbf{B}_0$ is used, which was found to be reliable for the other spacecraft. We note that THA plasma measurements were not available prior to 22:08 UT. Magnetosheath intervals have been removed from THA, THD and THE observations, identified when the electron density was greater than 5 cm$^{-3}$ or the magnetic field strength was less than 45 nT. Vectors within the magnetosphere have been rotated into local orthogonal field-aligned coordinates $(r, \phi, \parallel)$. The field-aligned direction $(\parallel)$ is given based on a robust linear regression of the magnetic field vectors[70,71], with the azimuthal $(\phi)$ direction being the cross product of $\parallel$ with the spacecraft's geocentric position thus pointing eastward, and the radial $(r)$ direction completing the right-handed set directed away from the Earth. While this coordinate rotation may result in some small $E_\parallel$, these are negligible compared to the other components and do not influence the results.

**Global MHD simulations.** We reproduce a high-resolution $(\frac{1}{8} - \frac{1}{16} R_E$ in the regions considered in this paper, see Supplementary Fig. 4 for grid) Space Weather Modeling Framework (SWMF[72,73];) simulation run of MSE excited by a 1 min solar wind density pulse (with sunward normal) under northward IMF[32]. Full details of the run are given in Supplementary Table 1. For all simulation quantities, perturbations are defined as the difference to the linear trend before $(t = 0$ min) and after $(t = 60$ min) the response to the pulse. Vectors are rotated into similar local field-aligned coordinates. The magnetopause location is determined as the last closed field line along geocentric rays through a bisection method accurate to

0.01 $R_E$. The bow shock standoff distance has been identified via interpolation as the point where the density is half that in the solar wind. In displaying perturbations in the simulation, a bi-symmetric log transform[74] is often employed due to the much larger amplitudes present during compression and rebound phases.

**Time-based filtering**. A time-based filtering technique is used to extract MSE wave perturbations and suppress noise and higher/lower frequency signals. This was chosen to avoid the potential influence of ringing artefacts or edge effects when using frequency-based methods due to the nonstationary process. Nonetheless, several different filtering methods were tested and the main results of the paper remained robust.

In the method presented for the spacecraft observations, first the raw data is smoothed using a 400 s robust LOESS method[75]. For stationary processes, this has a corresponding cutoff frequency of 3.6 mHz, therefore retains both the 1.8 mHz fundamental and 3.3 mHz second harmonic MSE signals present[35]. To remove any lower frequency trends still present, the mean envelope from cubic Hermite interpolation[76] is subtracted. These effectively bandpass filtered quantities are used for calculating the instantaneous wave Poynting vectors and energy densities. A time-averaging method is performed also by using the mean envelope from interpolation. The time-based methods used also allow uncertainties to be estimated. This is done via a running root-mean-squared (RMS) deviation between the raw and LOESS smoothed time-series, which are then propagated through the subsequent methods used[77,78].

To extract the MSE signal from the simulation, either in magnetopause location or grid point data, we firstly neglect the initial large amplitude compression and rebound. This is done by only using data from half an MSE period after the magnetopause's return to equilibrium, i.e. after the dotted line in Fig. 3a. For grid point data, the timing at the magnetopause with the same X coordinate is used. The secondary spectral peak is then suppressed using the same filtering procedure as for the THEMIS data. The only differences are that standard (rather than robust) LOESS is used due to reduced temporal resolution, and the window size used was 570 s corresponding to a 2.4 mHz cutoff.

**Fourier and wavelet techniques**. To compute time-averaged Poynting vectors as a function of frequency a standard complex Fourier approach is used (Eq. (2) of ref. [79])

$$\langle \mathbf{S}(\omega) \rangle = \frac{\Re e(\mathbf{E}(\omega) \times \mathbf{B}^*(\omega))}{2\mu_0} \quad (14)$$

This is done both in frequency–space using Welch's method[80] in computing one-sided cross spectra, i.e. the products of electric and magnetic field components, and in time–frequency space using products of analytic Morse continuous wavelet transforms[81]. In both cases, a null hypothesis of autoregressive noise is assumed, where the AR(1) parameters for each component of the electric and magnetic fields are estimated using constrained maximum likelihood and 500 independent Monte Carlo simulations are performed based on these models[82], with 95% confidence intervals being constructed by taking percentiles (2.5 and 97.5%) of the resulting time-averaged Poynting vectors.

**Slowness**. To quantify the propagation of MSE boundary perturbations, the slowness was computed through cross-correlating the filtered magnetopause signals between adjacent local times. By interpolating the peak to find its corresponding time lag $\Delta t$, the azimuthal slowness is given by

$$s_\phi = \frac{\Delta t}{|\Delta \mathbf{r}|} \quad (15)$$

where $|\Delta \mathbf{r}|$ is the distance between the two points on the boundary used. Standard errors in the correlation coefficient were also calculated and propagated through the interpolation procedure to arrive at uncertainties.

## Data availability

The THEMIS spacecraft data are available at http://themis.ssl.berkeley.edu/data/themis/ where level-2 data from the FGM, ESA and EFI instruments on each spacecraft has been used in this study. The SWMF simulation data generated in this study are available in the Community Coordinated Modeling Center (CCMC) at https://ccmc.gsfc.nasa.gov/results/viewrun.php?domain=GM&runnumber=Michael_Hartinger_061418_1.

## Code availability

The SWMF and BATS-R-US (Block-Adaptive Tree Solarwind Roe-type Upwind Scheme) software are available at https://github.com/MSTEM-QUDA. The SWMF and BATS-R-US tools used are available at https://ccmc.gsfc.nasa.gov.

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

## Acknowledgements

M.O.A. holds a UKRI (STFC/EPSRC) Stephen Hawking Fellowship EP/T01735X/1. M.D.H. was supported by NASA grant 80NSSC19K0127. F.P. was supported by the Austrian Science Fund (FWF): P 33285-N. D.J.S. was supported by STFC grant ST/S000364/1. We acknowledge NASA contract NAS5-02099 for use of data from the THEMIS Mission. Specifically K.H. Glassmeier, U. Auster and W. Baumjohann for the use of FGM data provided under the lead of the Technical University of Braunschweig and with financial support through the German Ministry for Economy and Technology and the German Center for Aviation and Space (DLR) under contract 50 OC 0302; C.W. Carlson and J.P. McFadden for use of ESA data; and J.W. Bonnell and F.S. Mozer for EFI data. Simulation results have been provided by the Community Coordinated Modeling Center (CCMC) at Goddard Space Flight Center using the SWMF and BATS-R-US tools developed at the University of Michigan's Center for Space Environment Modeling (CSEM).

## Author contributions

M.O.A. and M.D.H. conceived the study. L.R. gave technical support. M.O.A. performed the analysis of the data. M.O.A., M.D.H., F.P. and D.J.S. interpreted the results. M.O.A. wrote the paper, with M.D.H., F.P. and D.J.S. assisting.

## Competing interests

The authors declare no competing interests.
