## [Peer Review File · Nature Communications]

REVIEWER COMMENTS

Reviewer #1 (Remarks to the Author):

Review: Magnetopause ripples going against the flow form azimuthally stationary surface waves – Archer et al. – NCOMMS-21-08225-T

This paper combines observations, numerical simulations and mathematical analysis to propose the existence of an azimuthal standing mode structure of a magnetopause surface wave over central parts of the dayside magnetosphere (09-15 MLT). Further down the flanks the waves are swept tailward by the magnetosheath flow. This azimuthally standing/propagating nature would be an interesting facet of global magnetospheric wave structure and would enhance our understanding of the dayside solar wind/ magnetosphere interaction.

Overall, the paper is presented well, the figures are very polished and the language used is clear. The use of three approaches to tackle the problem is impressive and I commend the authors on a very interesting study. I have quite a few minor points, mostly about the interpretation of the simulations that I would like to see responses to. The subject matter is I believe of significant interest for publication in Nature Communications, after addressing the following points:

General Comments

1) The simulation movie provided is very helpful in assessing the simulation results, thanks for its inclusion. I do think however that there is potentially an over-reliance on the pure surface mode interpretation of these results. My biggest concern with the interpretation is there seems to be clear evidence of body/waveguide modes in the magnetosphere throughout the simulation, which are not given any credence in the text, despite the paper reporting to address the 'resonant response of the magnetospheric system globally' – line 51/52.

Below (attached as reviewer attachment - Archer_Screenshot.png) I've taken a screenshot from the movie provided, at T=21 minutes, after the passing of the initial impulse and reflection (as is indicated for the analysis in the text).

It seems clear to me that there is a non-evanescent radial structure across the dayside region of interest (09-15MLT), with peaks and troughs of significant amplitude, am I mistaken in this? It does seem that this could not be evaluated as having a radial structure described purely by a decaying exponential $\propto e^{-k_r r}$, which is assumed for much of the mathematical analysis and interpretation which follows (e.g. I presume in the derivation of equation (8) using surface mode form from Plaschke and Glassmeier 2011.)

It seems to me that the system needs treated as a combination of these approaches (waveguide/surface). The 'normal mode' of the magnetosphere contains all of this structure, both within the magnetosphere and on the magnetopause boundary, and given that it's all one system I'm not sure that it can be so easily separated.

Similar features to the simulations shown in this manuscript were presented in simulations by Elsden and Wright, 2019 (see reference list at end). They showed that the presence of an inner reflecting boundary/turning point (e.g. plasmopause) set up an azimuthally standing mode structure around noon, and a propagating nature on the flanks. This was shown to occur when the Alfvén speed decreased with azimuth, which occurs naturally in the magnetospheric waveguide because of the flaring of the flanks i.e. $V_A(L=10)$ at noon $>$ $V_A(L=15)$ on the flanks. Such normal mode structure was also shown in a 3D dipole magnetospheric model with flared flanks (Wright and Elsden, 2020). Therefore, I wonder if the authors could comment on the behaviour inside of the boundary, the potential presence of waveguide modes in the simulations and how these may affect/add to the interpretation of the simulations. Plots of the radial variation of b_{\parallel} along say the noon meridian at different times in the simulation, may help with this.

2) In figure 1, there's an interesting difference between the THA and THE Poynting vector/ energy velocity signals. In THA, the averaged azimuthal Poynting vector (panel vii) turns positive (Eastward) at ~ 2235 UT and remains so for the rest of the time window displayed. At THE, at the end of the time window, the signal turns significantly azimuthally Westward (2245-2250 panel vii), suggesting changing radial structure of the signal given the different radial locations of the spacecraft (approx 2RE apart). Therefore the time window of positive Eastward azimuthal Poynting vector at both spacecraft is only about 10 minutes, which is approximately one wave period of the proposed MSE frequency (1.8mHz – maybe mention this frequency within the main Spacecraft observations section as well as the Methods section?). Could the authors please comment on potential explanations for this reasonably small scale radial structuring within the surface mode description as well as the implications of the short duration of observation of +ve S_{ϕ} relative to the wave period?

3) I am a little confused at the reasoning following equation (5) regarding small real k_r (lines 85-88). Surely this argument requires a balancing of the terms in equation (5), and will vary depending upon the values of the frequency and wavenumbers? For instance, for small k_{ϕ} one would expect a less evanescent radial structure etc. It also took me a little while to see that complex frequency ω

necessitates a real part to k_r , so perhaps this could be outlined more explicitly. A little extra discussion on the potential sizes of the terms would be useful.

4) In the methods section on global MHD simulations, line 265, could you please rephrase 'high-resolution (up to $1/16$ RE)' when referring to the simulations as it seems to me to be rather misleading. Checking Table 2, this resolution is only used over $3 < r < 5$, and since the entirety of the interest/analysis in this paper is done outside this region, where the resolution is only $1/4$ RE, this is not an accurate representation of the run. I would also therefore ask if the magnetopause was sufficiently resolved in this study? Given that there's only 8 grid points across $2RE$, I would start to worry about sufficient resolution of the mode structure, in particular the spatial interpretation of second order quantities like the Poynting vector. Did the authors perform the same analysis at different resolutions to check for consistency between runs?

5) When referencing which paper an equation is from, could the authors please reference the equation numbers in those studies too please? This makes looking up these papers for information much easier. For example, equation (6) apparently based on Junginger, 1985, is this after equation (15) in that paper for their S_z ? Equally line 184/185 using the surface wave dispersion relation of [9] (Plaschke and Glassmeier 2011), it would be helpful to include this extra line of algebra. Furthermore, could you please add in the theory description in lines 177-181 that it is a homogeneous plasma you are considering? Well I presume this to be the case following the analysis of Junginger, 1985.

Minor Comments

1. Fig1 panel (ix)– perhaps add that you've plotted the absolute value of the magnetosheath flow speed in grey? I found this confusing when in the text you referred to the Poynting vector azimuthal component as being oppositely directed (line 77), yet they occupy the same upper half of the graph.

2. Reference 18 Junginger not Juninger...

3. Typo line 188 – on the where...

4. Fig 4 caption – mis-label on figures b) and c) for large and small.

5. Line 231 – behind its

References

Elsden, T. and Wright, A. N. (2019), The Effect of Fast Normal Mode Structure and Magnetopause Forcing on FLRs in a 3D Waveguide, *J. Geophys. Res. Space Physics*, 123, doi:10.1029/2018JA026222

Wright, A. N., & Elsden, T. (2020). Simulations of MHD wave propagation and coupling in a 3-D magnetosphere. *Journal of Geophysical Research: Space Physics*, 125, e2019JA027589., doi:10.1029/2019JA027589

Reviewer #2 (Remarks to the Author):

Review of the manuscript Magnetopause ripples going against the flow form azimuthally stationary surface waves by M. O. Archer et al.

The manuscript presents the results of multi-spacecraft observations, global simulations, and analytic theory to claim that azimuthally stationary magnetopause surface waves may occur between 09–15h magnetic local time with the waves propagating against the magnetosheath flow, exactly balancing its advective effect leading to no net energy flux.

The standing magnetopause waves are discussed usually in terms of cavity modes and/or the surface magnetopause modes. I think some discussion presenting differences of these modes (which can be used to distinguish them) can be useful.

The comprehensive numerical model nicely represents the idea of the proposed scenario (in Figure 3). The movie would benefit if the authors highlight the regions, where the discussed phenomenon can be seen (probably, the log scale for the magnetic field perturbation can be helpful) – the tailward propagation dominates at the moment.

I would propose to find a better experimental example: only ~10 minutes of the discussed interval can be used as a confirmation of the model but this is about only one period of the discussed wave. The Poynting flux directions before and after contradict the model. The intensive filtering of the data can lead to a false wave appearance - probably, presenting of the wavelet-based full picture could be more suitable. The phase shift between THA and THE is not discussed but could be an argument for the model if matched it (but looks like it doesn't). The field-aligned electric field measurements need to be clarified (the source and quality).

I think that the paper needs a revision of the experimental part to be reconsidered for publication.

Reviewer #3 (Remarks to the Author):

Overview: The papers shows very nice multi-spacecraft

observations, global simulations, and uses analytic theory to investigate properties (e.g., Poynting vector) of magnetopause surface waves. The authors show that the azimuthally stationary magnetopause

surface waves may occur between 09–15h magnetic local time.

I find the paper easy to read and well written and I agree with the analysis and the results.

However, I somewhat disagree with the authors statement that the stationarity of the surface waves has been a mystery as it is expected that they move tailward with magnetosheath flow.

Magnetosheath is governed by sub-magnetosonic plasma flow so when the surface waves are driven by magnetosonic waves (as authors state in equation 5) it is not surprising at all that they could propagate upstream balancing the tailward advection and thus remain relatively stationary with respect to magnetopause at 9-15 h.

Recommendation: I think the paper is an important contribution and ready for publication as is in a more narrowly focused journal like JGR-Space Physics. However, I think that for Nature Communication (with broad audience) the authors should try make the motivation of the work more compelling from the beginning (abstract). For example, the statement in the line 12, as stated in the abstract, is a not a 'surprising mystery' to me (but this may depend on the reader). Describe more

what are the important implications of these results for global magnetospheric dynamics and fields/areas outside magnetospheric physics where surface waves can form.

Response to reviewers

'Magnetopause ripples going against the flow form azimuthally stationary surface waves' Archer et al.

We would like to thank all the reviewers for their thoughtful comments in assessing our manuscript. We have considered them thoroughly and in making revisions believe they have strengthened the study. Please find our responses below, where line numbers refer to the tracked changes version of the revised manuscript.

Reviewer #1

Review: Magnetopause ripples going against the flow form azimuthally stationary surface waves – Archer et al. – NCOMMS-21-08225-T

This paper combines observations, numerical simulations and mathematical analysis to propose the existence of an azimuthal standing mode structure of a magnetopause surface wave over central parts of the dayside magnetosphere (09-15 MLT). Further down the flanks the waves are swept tailward by the magnetosheath flow. This azimuthally standing/propagating nature would be an interesting facet of global magnetospheric wave structure and would enhance our understanding of the dayside solar wind/ magnetosphere interaction.

Overall, the paper is presented well, the figures are very polished and the language used is clear. The use of three approaches to tackle the problem is impressive and I commend the authors on a very interesting study. I have quite a few minor points, mostly about the interpretation of the simulations that I would like to see responses to. The subject matter is I believe of significant interest for publication in Nature Communications, after addressing the following points:

General Comments

1) The simulation movie provided is very helpful in assessing the simulation results, thanks for its inclusion. I do think however that there is potentially an over-reliance on the pure surface mode interpretation of these results. My biggest concern with the interpretation is there seems to be clear evidence of body/waveguide modes in the magnetosphere throughout the simulation, which are not given any credence in the text, despite the paper reporting to address the 'resonant response of the magnetospheric system globally' – line 51/52.

We thank the reviewer for this comment and agree that discussion of body (and in particular waveguide) modes would improve the manuscript. Mention of these has body waves has been added to the introduction on lines 58-64.

Previous publications have compared in detail observations (Archer et al., 2019) and simulations (Hartinger et al., 2015) of the response on the dayside magnetosphere to expectations for both surface and waveguide modes, finding them consistent only with the surface mode interpretation. Due to the word limits of the journal we do not reproduce these arguments in full in this manuscript but simply highlight the outcomes. The strongest argument against a waveguide interpretation on the dayside is on the basis of frequency – waveguide modes on the dayside cannot occupy the same low frequency range as the surface mode, which we have expanded discussion of on lines 75-77 and 223-225.

We do note though that we have indeed found evidence of the surface eigenmode coupling to a waveguide mode in the simulation at the terminator, where the frequencies of these two modes become similar. This is now discussed on lines 250-258.

Through these revisions we feel the discussion now better reflects the resonant response of the magnetospheric system globally.

Archer, M.O., Hietala, H., Hartinger, M.D. et al. Direct observations of a surface eigenmode of the dayside magnetopause. Nat Commun 10, 615 (2019). <https://doi.org/10.1038/s41467-018-08134-5>

Hartinger, MD, Plaschke, F, Archer, MO, Welling, DT, Moldwin, MB, and Ridley, A (2015), The global structure and time evolution of dayside magnetopause surface eigenmodes. Geophys. Res. Lett., 42, 2594– 2602. doi: 10.1002/2015GL063623.

Below (attached as reviewer attachment - Archer_Screenshot.png) I've taken a screenshot from the movie provided, at T=21 minutes, after the passing of the initial impulse and reflection (as is indicated for the analysis in the text).

It seems clear to me that there is a non-evanescent radial structure across the dayside region of interest (09-15MLT), with peaks and troughs of significant amplitude, am I mistaken in this? It does seem that this could not be evaluated as having a radial structure described purely by a decaying exponential $[\propto e^{-k_r r}]$, which is assumed for much of the mathematical analysis and interpretation which follows (e.g. I presume in the derivation of equation (8) using surface mode form from Plaschke and Glassmeier 2011.)

The non-evanescent radial structure that the reviewer refers to is due purely to radial propagation of phase fronts towards the magnetopause on the magnetospheric side (i.e. a complex k_r). This was discussed in the manuscript as being a consequence of the damping of the surface modes. The added Figure 4a now clearly shows this slow (30-40 km/s) propagation of phase fronts whilst also revealing that the amplitudes of magnetic and velocity perturbations do indeed decay with distance from the magnetopause on the magnetospheric side, as expected for a surface eigenmode. This is now explained more fully on lines 229-233.

It seems to me that the system needs treated as a combination of these approaches (waveguide/surface). The 'normal mode' of the magnetosphere contains all of this structure, both within the magnetosphere and on the magnetopause boundary, and given that it's all one system I'm not sure that it can be so easily separated.

We agree with the reviewer that the waveguide modes also constitute a normal mode of the magnetospheric system. Our revised introduction includes more discussion of waveguide modes in this context. We also agree with the reviewer that the surface eigenmode may couple to the waveguide mode in regions of the magnetosphere where their frequencies are sufficiently similar. While at the subsolar magnetosphere the expected quarter wavelength mode waveguide frequency will typically be much higher than the surface eigenfrequency (e.g. it is around 4 mHz in the simulation), further down the flanks this coupling is more likely to be possible due to the larger cavity and lower wave speeds. Indeed, in the added Figure 4b we show that the seeded tailward propagating surface waves with a 10 min periodicity are able to couple to quarter wavelength waveguide mode at the terminator. The following discussion on lines 254-257 therefore discuss the possibility of this combined normal mode.

Similar features to the simulations shown in this manuscript were presented in simulations by Elsden and Wright, 2019 (see reference list at end). They showed that the presence of an inner reflecting boundary/turning point (e.g. plasmopause) set up an azimuthally standing mode structure around noon, and a propagating nature on the flanks. This was shown to occur when the Alfvén speed decreased with azimuth, which occurs naturally in the magnetospheric waveguide because of the flaring of the flanks i.e. $V_A(L=10)$ at noon $>$ $V_A(L=15)$ on the flanks. Such normal mode structure was also shown in a 3D dipole magnetospheric model with flared flanks (Wright and Elsden, 2020). Therefore, I wonder if the authors could comment on the behaviour inside of the boundary, the potential presence of waveguide modes in the simulations and how these may affect/add to the interpretation of the simulations. Plots of the radial variation of b_{\parallel} along say the noon meridian at different times in the simulation, may help with this.

We thank the reviewer for these comments, which we believe strengthen our interpretation of the results on the dayside magnetosphere as being further evidence of the surface eigenmode and we have added the reviewer's suggested plots as Figure 4.

The fundamental frequency in Wright & Elsden (2020) is 3.9 mHz, in good agreement with a quarter wavelength mode based on the travel time. The travel time on the dayside in our simulation is similar and yet the waves we observe are 1.4 mHz indicating a different mode entirely. The waveguide simulations all have strong compressional perturbations at the inner boundary, whereas ours are strongest in amplitude close to the magnetopause.

We note, however, that our simulations show agreement with these two points along the terminator, where we present evidence that the surface eigenmode couples to a waveguide mode in a region where their frequencies become similar.

Finally, the scenario for an azimuthally stationary fast magnetosonic wave on the dayside proposed in Elsden and Wright (2019) should result in zero time-averaged Poynting flux azimuthally. In contrast, we both observe and simulate resultant Poynting fluxes that are in opposition to and balancing the magnetosheath flow (something which the simulations the reviewer refers to does not include).

2) In figure 1, there's an interesting difference between the THA and THE Poynting vector/ energy velocity signals. In THA, the averaged azimuthal Poynting vector (panel vii) turns positive (Eastward) at ~ 2235 UT and remains so for the rest of the time window displayed. At THE, at the end of the time window, the signal turns significantly azimuthally Westward (2245-2250 panel vii), suggesting changing radial structure of the signal given the different radial locations of the spacecraft (approx 2RE apart). Therefore the time window of positive Eastward azimuthal Poynting vector at both spacecraft is only about 10 minutes, which is approximately one wave period of the proposed MSE frequency (1.8mHz – maybe mention this frequency within the main Spacecraft observations section as well as the Methods section?). Could the authors please comment on potential explanations for this reasonably small scale radial structuring within the surface mode description as well as the implications of the short duration of observation of +ve S_{ϕ} relative to the wave period?

At 22:45 another jet impinged on the magnetopause. We now show the driving pressure as panel i of Figure 1b to indicate this along with the observed magnetopause motion as panel ii. In this paper we are interested in the response during the period of little direct driving on the magnetopause by total pressure variations, indicated by the added vertical dotted lines. During this period only two oscillations of the boundary occurred. Magnetopause surface modes are thought to be strongly damped, a point which we now emphasise in the manuscript, which is why the energy flow towards the subsolar point occurs only over a short period.

The characteristic transverse scale sizes of magnetosheath jets are thought to be around 0.1 RE (Plaschke et al., 2020), therefore it is not surprising that the initial transient response to the second jet results in differences between the spacecraft, given that THA and THE are separated by approximately 1 RE tangential to the magnetopause. However, it is the resonant response which follows the jets that is of interest in this paper and we find that after this second jet both spacecraft again observe eastward Poynting fluxes.

Indeed, following several other jets on this day we observed Poynting vectors with the same sense as the interval presented. Therefore, we now show as Figure 2 the overall direction of the time-averaged Poynting vector for a wider interval using Fourier methods. This also shows agreement with the direction presented in the focused interval.

Plaschke, F., Hietala, H., & Vörös, Z. (2020). Scale sizes of magnetosheath jets. *Journal of Geophysical Research: Space Physics*, 125, e2020JA027962.
<https://doi.org/10.1029/2020JA027962>

3) I am a little confused at the reasoning following equation (5) regarding small real k_r (lines 85-88). Surely this argument requires a balancing of the terms in equation (5), and will vary depending upon the values of the frequency and wavenumbers? For instance, for small k_ϕ one would expect a less evanescent radial structure etc. It also took me a little while to see that complex frequency ω necessitates a real part to k_r , so perhaps this could be outlined more explicitly. A little extra discussion on the potential sizes of the terms would be useful.

We thank the reviewer for these comments and in response have expanded the discussion as to why wave damping results in a small real k_r , which can be found on lines 142-147. While the reviewer is correct that k_r does indeed depend on the particular values used, on the magnetospheric side of the boundary the low plasma beta simplifies the third term of the equation considerably. We have also calculated typical radial phase velocities due to this effect, showing them to be consistent with both the observations (lines 147-150) and simulation (lines 230-232).

4) In the methods section on global MHD simulations, line 265, could you please rephrase 'high-resolution (up to 1/16 RE)' when referring to the simulations as it seems to me to be rather misleading. Checking Table 2, this resolution is only used over $3 < r < 5$, and since the entirety of the interest/analysis in this paper is done outside this region, where the resolution is only 1/4 RE, this is not an accurate representation of the run. I would also therefore ask if the magnetopause was sufficiently resolved in this study? Given that there's only 8 grid points across 2RE, I would start to worry about sufficient resolution of the mode structure, in particular the spatial interpretation of second order quantities like the Poynting vector. Did the authors perform the same analysis at different resolutions to check for consistency between runs?

The reviewer makes a valid point that only referring to the 1/16 RE resolution may be misleading. However, the reviewer is mistaken that the resolution of the main region of interest was only 1/4 RE, as it was 1/8 RE. Therefore, the magnetopause boundary was sufficiently resolved, given its 1.5 RE width in the simulation. Hartinger et al. (2015) investigated how grid resolution affects the results, finding it simply changed the damping rate present in the simulations due to numerical dissipation. As we acknowledge that the grid resolutions as listed in Table 2 may be difficult to interpret due to the many different regions, we have added a supplementary figure as well which more clearly depicts the different regions of grid resolutions used.

Hartinger, MD, Plaschke, F, Archer, MO, Welling, DT, Moldwin, MB, and Ridley, A (2015), The global structure and time evolution of dayside magnetopause surface eigenmodes. *Geophys. Res. Lett.*, 42, 2594– 2602. doi: 10.1002/2015GL063623.

5) When referencing which paper an equation is from, could the authors please reference the equation numbers in those studies too please? This makes looking up these papers for information much easier. For example, equation (6) apparently based on Junginger, 1985, is this after equation (15) in that paper for their S_z ? Equally line 184/185 using the surface wave dispersion relation of [9] (Plaschke and Glassmeier 2011), it would be helpful to include this extra line of algebra. Furthermore, could you please add in the theory description in lines 177-181 that it is a homogeneous plasma you are considering? Well I presume this to be the case following the analysis of Junginger, 1985.

We have made all of these changes.

Minor Comments

1. Fig1 panel (ix)– perhaps add that you’ve plotted the absolute value of the magnetosheath flow speed in grey? I found this confusing when in the text you referred to the Poynting vector azimuthal component as being oppositely directed (line 77), yet they occupy the same upper half of the graph.

We have made this change.

2. Reference 18 Junginger not Juninger...

Thank you for spotting this mistake.

3. Typo line 188 – on the where...

We have corrected this.

4. Fig 4 caption – mis-label on figures b) and c) for large and small.

We have corrected this.

5. Line 231 – behind its

We have corrected this.

References

Elsden, T. and Wright, A. N. (2019), The Effect of Fast Normal Mode Structure and Magnetopause Forcing on FLRs in a 3D Waveguide, *J. Geophys. Res. Space Physics*, 123, doi:10.1029/2018JA026222

Wright, A. N., & Elsden, T. (2020). Simulations of MHD wave propagation and coupling in a 3-D magnetosphere. *Journal of Geophysical Research: Space Physics*, 125, e2019JA027589., doi:10.1029/2019JA027589

Reviewer #2

Review of the manuscript Magnetopause ripples going against the flow form azimuthally stationary surface waves by M. O. Archer et al.

The manuscript presents the results of multi-spacecraft observations, global simulations, and analytic theory to claim that azimuthally stationary magnetopause surface waves may occur between 09–15h magnetic local time with the waves propagating against the magnetosheath flow, exactly balancing its advective effect leading to no net energy flux.

The standing magnetopause waves are discussed usually in terms of cavity modes and/or the surface magnetopause modes. I think some discussion presenting differences of these modes (which can be used to distinguish them) can be useful.

We agree with the reviewer that discussing cavity/waveguide modes would improve the manuscript and we have added this to the introduction.

The comprehensive numerical model nicely represents the idea of the proposed scenario (in Figure 3). The movie would benefit if the authors highlight the regions, where the discussed phenomenon can be seen (probably, the log scale for the magnetic field perturbation can be helpful) – the tailward propagation dominates at the moment.

We have changed the colour scale in the movie to a bi-symmetric log scale and added dotted lines indicating hours of local time to the XY plane.

I would propose to find a better experimental example: only ~10 minutes of the discussed interval can be used as a confirmation of the model but this is about only one period of the discussed wave.

At present only one confirmed interval of a magnetopause surface eigenmode has been published. This is because, as detailed in Archer et al. (2019), simultaneous observations of an isolated impulsive event (magnetosheath jet), motion of the magnetopause boundary, and magnetospheric ULF wave signatures by at least four spacecraft in a string-of-pearls configuration were required for confirmation. These are very strict constraints, hence the same event is employed in this study. However, we stress that the observations are simply leveraged to pose a hypothesis around azimuthal propagation. It is through the simulations and theory that this hypothesis is more fully explored, due to the challenges associated with observing this mode.

Magnetopause surface modes are thought to be strongly damped, a point which we now emphasise in the manuscript, which is why the energy flow towards the subsolar point occurs only over a short interval. Indeed only two oscillations of the magnetopause were observed by the spacecraft following the first jet. We now show the driving pressure as panel i of Figure 1b to indicate this along with the observed magnetopause motion as panel ii to provide readers with more context of this event.

We also now note in the text that several other instances of Poynting fluxes directed azimuthally eastward following jets were observed. To demonstrate this the time-averaged Poynting vector as a function of frequency is presented for a much wider interval as the added Figure 2. This clearly reproduces the radially outwards and azimuthally eastward direction, being a statistically significant result.

Archer, M.O., Hietala, H., Hartinger, M.D. et al. Direct observations of a surface eigenmode of the dayside magnetopause. *Nat Commun* 10, 615 (2019). <https://doi.org/10.1038/s41467-018-08134-5>

The Poynting flux directions before and after contradict the model.

In this paper we are interested in the resonant response of the magnetopause during the period of little direct driving by total pressure variations, indicated by the added vertical dotted lines. Directly before and after these times there were magnetosheath jets impinging on the boundary (see panel i of Figure 1b) which will initially excite a direct response due to the pressure imbalance and then lead to the excitation of surface eigenmodes.

The intensive filtering of the data can lead to a false wave appearance - probably, presenting of the wavelet-based full picture could be more suitable.

We chose to use time-based smoothing and detrending in the paper because it is the simplest method to understand. Each step of the filtering from raw data to final perturbations used in calculating the Poynting flux and energy velocity are explicitly shown in Figure 1 so that readers can follow the processing. Furthermore, uncertainties resulting from the filtering can be easily estimated using time-based methods, which we have done and shown to indicate the likely size of potential spurious signals. Frequency-based methods, in contrast, can be subject to ringing artefacts and edge effects when dealing with nonstationary processes and robust uncertainty estimation associated with these can be challenging.

Nonetheless, we tried several different filtering techniques to extract the wave perturbations before settling on the time-based method. These included wavelet filtering of the entire interval, or selecting only the period between the jets (dotted lines) and then filtering using wavelet and Fourier techniques. We found the main results of the paper were maintained, as we now note on lines 434-435.

We now also include a Fourier analysis of a wider interval of THA data to lend further confidence to the result. These are shown as the added Figure 2. The radial and azimuthal Poynting vector components for this 82 minute interval were found to be statistically significant at the MSE frequencies and directed in agreement with the time-based method. This is discussed on lines 173-182.

The phase shift between THA and THE is not discussed but could be an argument for the model if matched it (but looks like it doesn't).

All three spacecraft (THA, THD, THE) show excellent agreement in the radial and azimuthal directions as to their direction and magnitude of energy flux/velocity. It is these directions that are the focus of the paper. We have, however, expanded discussion of the energy flux in the field-aligned direction on lines 156-171 explaining why some differences between THA and THD/THE might actually be expected.

The field-aligned electric field measurements need to be clarified (the source and quality).

While direct measurements of the parallel electric field are available from the axial boom of the THEMIS EFI instrument, they are only reliable at much higher frequencies than considered here, thus we replace the axial field using the $E \cdot B = 0$ approximation, which is standard at the ULF timescales (Bonnell et al., 2008). This is performed for THD and THE at each datapoint. The coordinate system definition, however, uses a longer-term average to determine the field-aligned

direction. The result is that there may be a negligible field-aligned electric field following the transformation. We stress though that this component does not contribute to the final results. For THA there were no EFI measurements available, hence we use velocity measurements ($E = -v \times B_0$) and so the field-aligned electric field is necessarily zero. Both velocity and electric field measurements agreed well for THD and THE, where both measurements were possible. These points are noted on lines 420-423.

Bonnell, J.W., Mozer, F.S., Delory, G.T. et al. The Electric Field Instrument (EFI) for THEMIS. *Space Sci Rev* 141, 303–341 (2008). <https://doi.org/10.1007/s11214-008-9469-2>

I think that the paper needs a revision of the experimental part to be reconsidered for publication.

We reiterate that the spacecraft observations are leveraged to pose a hypothesis which is then fully examined through simulations and theory. We hope that our clarifications and revisions have addressed the reviewer's concerns.

Reviewer #3

Overview: The papers shows very nice multi-spacecraft observations, global simulations, and uses analytic theory to investigate properties (e.g., Poynting vector) of magnetopause surface waves. The authors show that the azimuthally stationary magnetopause surface waves may occur between 09–15h magnetic local time.

I find the paper easy to read and well written and I agree with the analysis and the results.

However, I somewhat disagree with the authors statement that the stationarity of the surface waves has been a mystery as it is expected that they move tailward with magnetosheath flow.

Magnetosheath is governed by sub-magnetosonic plasma flow so when the surface waves are driven by magnetosonic waves (as authors state in equation 5) it is not surprising at all that they could propagate upstream balancing the tailward advection and thus remain relatively stationary with respect to magnetopause at 9-15 h.

We take the reviewer's point and have reframed the abstract, introduction, and much of the main text in response. In magnetospheric dynamics, current models for external driving (even under impulsive excitation as discussed in this manuscript) predict the tailward propagation of disturbances and their associated waves across the entire outer magnetosphere (e.g. Sibeck, 1990). This is because the driving pressure fronts and/or shear flows travel in this direction, imparting their momentum via surface waves into geospace. Many reviews of ULF wave observations conform to this paradigm (e.g. Keiling et al., 2016). Instances of sunward propagating ULF waves have usually been thought to be caused internally by energetic particle instabilities (e.g. Constantinescu et al., 2009) or due to changes in the tail (Nielsen, 1984; Eriksson et al., 2008). Therefore, the results in this paper challenge a well-accepted paradigm in global magnetospheric dynamics.

We also point out that the submagnetosonic magnetosheath flow is not sufficient at fully explaining the results, as this is true between 07.6-16.4h MLT yet the azimuthally stationary surface waves are constrained to a smaller region than this. The 'Analytic MHD theory' section, however, well describes the region of stationarity.

E. Nielsen. Observations of sunward propagating waves on the magnetopause. *J. Geophys. Res.*, 89:9095–9099, 1984.

P. T. I. Eriksson, L. G. Blomberg, S. Schaefer, and K.-H. Glassmeier. Sunward propagating Pc5 waves observed on the post-midnight magnetospheric flank. *Ann. Geophys.*, 26:1567–1579, 2008.

O. D. Constantinescu, K.-H. Glassmeier, F. Plaschke, U. Auster, V. Angelopoulos, W. Baumjohann, K.-H. Fornçon, E. Georgescu, D. Larson, W. Magnes, J. P. McFadden, R. Nakamura, and Y. Narita. THEMIS observations of duskside compressional Pc5 waves. *J. Geophys. Res.*, 114:A00C25, 2009.

A. Keiling, D.-H. Lee, and V. Nakariakov, editors. *Low-Frequency Waves in Space Plasmas*. Geophysical Monograph Series. American Geophysical Union, 2016.

Recommendation: I think the paper is an important contribution and ready for publication as is in a more narrowly focused journal like *JGR-Space Physics*. However, I think that for *Nature Communication* (with broad audience) the authors should try make the motivation of the work more compelling from the beginning (abstract). For example, the statement in the line 12, as stated in the abstract, is a not a ‘surprising mystery’ to me (but this may depend on the reader). Describe more what are the important implications of these results for global magnetospheric dynamics and fields/areas outside magnetospheric physics where surface waves can form.

We also take the reviewer’s point on the importance of this work. In the introduction we now discuss the importance of surface wave mechanisms to many space, astrophysical and laboratory plasma systems. Earth’s magnetosphere is therefore an accessible laboratory for *in situ* measurements of these processes, with applications to other environments such as coronal loops whose oscillations share many conceptual similarities to surface eigenmodes. We have expanded the broad implications of the results in the Discussion on both global magnetospheric dynamics at Earth (e.g. radial diffusion in the radiation belts, and auroral, ionospheric, and ground-based magnetometer measurements), other planetary magnetospheres, and areas outside of magnetospheric physics where surface waves can form.

REVIEWER COMMENTS

Reviewer #1 (Remarks to the Author):

The authors have addressed many of my comments from the initial review and the manuscript, I believe, has improved substantially. With the introduction of a couple of new figures I have a few further minor comments to be addressed.

1) Figure 2 – what is the explanation for the parallel Poynting vector being positive for low frequencies?

2) Thanks for including the grid resolution as a supporting figure. I would suggest that you could lose the confusing grid in Table 2 but I'll leave that up to you. I see now that my confusion came from the fact there are overlapping ranges in x, y, z, which should be changed as it's unclear which resolution applies when the same region is listed twice (this is made obvious from the figure!)

3) Something which I would like some comments on is the suitability of comparison between the simulation and observations given the substantially different driving mechanism used. When replying to a previous comment of mine you mentioned the transverse scale sizes of the jet as being responsible for the differences between THA and THE signals given they're 1RE apart. However, in the simulation, you are driving with a pressure pulse with an azimuthal scale length of the size of the entire magnetosphere. I realise that more specific local drivers cannot be placed within the global MHD simulations and so understand why you have done this, but a discussion on the relevance (and caveats) of this simulation based on the reality of local jet driving with small azimuthal scales should be included and argued for.

Reviewer #2 (Remarks to the Author):

The topic looks to me not to be general enough for Nature Communications. The manuscript is very well written and ready to be recommended for publication for a more specific journal (GRL or JGR).

The experimental case is not completely convincing and the authors' statement that this is a unique observation makes it interesting but unclear from point of view of its significance and potential importance for the system.

The filtering applied doesn't allow one to understand the dynamics of the system - I think the presentation in the time-frequency domain can be more helpful here. I didn't understand the answer to my comment about the observed phase shift at THAA and THE - this could be clear if the authors present virtual spacecraft time-series from two points corresponding to the THEMIS spacecraft locations.

I appreciate the revision of the movie presentation but it is still hard to get the point - I suggest highlighting the area and time supporting the authors' conclusion. Probably, as the system is symmetrical, only half might be presented to provide better resolution. The zoom of the area of interest with highlighting the phenomenon discussed can be helpful here.

Reviewer #3 (Remarks to the Author):

The authors have addressed all my comments and have greatly improved the manuscript by putting the new results in the context of the global magnetospheric dynamics. I can therefore recommend it for publication in Nature Communications after a final, minor suggestion:

To further emphasize importance for global magnetospheric dynamics and for global plasma circulation related to surface wave growth, my only remaining suggestion is to add somewhere, when discussing magnetospheric impacts, on lines 37 to 44 the following references to the first works demonstrating a physical mechanism for mass transport at the magnetopause produced by the non-linear KH waves (reconnection in KH vortices) which could grow from surface waves when sufficient free energy is present to overcome magnetic tension or plasma compressibility:

1. Nykyri, K. and Otto, A. (2001), Plasma transport at the magnetospheric boundary due to reconnection in Kelvin-Helmholtz vortices. *Geophys. Res. Lett.*, 28: 3565-3568.
<https://doi.org/10.1029/2001GL013239>

2. Ma, X., Otto, A., and Delamere, P. A. (2014), Interaction of magnetic reconnection and Kelvin-Helmholtz modes for large magnetic shear: 1. Kelvin-Helmholtz trigger, *J. Geophys. Res. Space Physics*, 119, 781– 797, doi:10.1002/2013JA019224.

Response to reviewers

Revision of 'Magnetopause ripples going against the flow form azimuthally stationary surface waves' Archer et al.

We would like to thank the reviewers for their comments on our revised manuscript. Please find our responses below, where line numbers refer to the tracked changes version of the new revision.

Reviewer #1

The authors have addressed many of my comments from the initial review and the manuscript, I believe, has improved substantially. With the introduction of a couple of new figures I have a few further minor comments to be addressed.

We thank the reviewer for their assessment that the revisions have improved the manuscript.

1) Figure 2 – what is the explanation for the parallel Poynting vector being positive for low frequencies?

The parallel Poynting flux is positive at MSE frequencies indicates that THA was overall located above the “null point” of the waves, which forms due to the asymmetries present, as well described by Allan (1982). We have added a note of this on lines 185-187.

2) Thanks for including the grid resolution as a supporting figure. I would suggest that you could lose the confusing grid in Table 2 but I'll leave that up to you. I see now that my confusion came from the fact there are overlapping ranges in x, y, z, which should be changed as it's unclear which resolution applies when the same region is listed twice (this is made obvious from the figure!)

We agree and have removed the lines from the table entirely.

3) Something which I would like some comments on is the suitability of comparison between the simulation and observations given the substantially different driving mechanism used. When replying to a previous comment of mine you mentioned the transverse scale sizes of the jet as being responsible for the differences between THA and THE signals given they're 1RE apart. However, in the simulation, you are driving with a pressure pulse with an azimuthal scale length of the size of the entire magnetosphere. I realise that more specific local drivers cannot be placed within the global MHD simulations and so understand why you have done this, but a discussion on the relevance (and caveats) of this simulation based on the reality of local jet driving with small azimuthal scales should be included and argued for.

Figure 6 highlights that, at each location on the boundary, after the other (blue) wavevectors have been swept downtail and the boundary has formed its resonance (pink wavevector), the physics of the azimuthally stationary surface wave is confined to a small local time region, i.e. a single meridian of geomagnetic field lines. While the initial perturbation on the boundary and the corresponding immediate transient response will depend on the specifics of the driving pressure variation (scale sizes, location of impact etc.), one can simply decompose the initial perturbation at each local time into the normal modes along the field (the surface eigenmodes) and this should entirely dictate the subsequent resonant response at that local time. Indeed, the local boundary motion and Poynting fluxes were in agreement across both observations and simulations despite different scale size drivers being leveraged. This locality to the physics means that the azimuthally

stationary wave should be limited to the local times in which the driver impacted the magnetopause, hence the scale of the driver in azimuth (within the 09–15h local time range) would be imprinted in the stationary waves excited. We have added this discussion on lines 343-353.

Reviewer #2

The topic looks to me not to be general enough for Nature Communications. The manuscript is very well written and ready to be recommended for publication for a more specific journal (GRL or JGR).

Surface waves serve as an important mechanism in filtering, accumulating, and guiding the disturbances between/through space, astrophysical and laboratory plasmas. They have been observed and modelled at tokamaks, planetary plasma tori, coronal loops, accretion disks, astrophysical jets to name a few. This makes understanding surface waves of universal importance, which we have stated in the introduction. Earth's magnetosphere provides the opportunity of studying surface wave processes *in situ* rather than through remote sensing. Our paper concerns magnetopause surface waves which are found not to conform with the well-established paradigm of tailward propagation in magnetospheric dynamics. However, we have broadened the discussion of this paradigm further in the introduction by analogy with the formation mechanisms of surface waves on water, which too propagate with the direction of the wind (lines 35-37). We believe the topic of surface waves and the results of waves going against the typical direction of propagation to therefore be of broad interest.

The experimental case is not completely convincing and the authors' statement that this is a unique observation makes it interesting but unclear from point of view of its significance and potential importance for the system.

We apologise that we were unclear in our response. The driving conditions themselves are not unique, magnetosheath jets are common impulsive drivers that impact the magnetopause regularly - 3 per hour in general and 9 per hour under low interplanetary magnetic field cone angle conditions (Plaschke et al., 2016). Several other possible drivers of surface eigenmodes also exist, including solar wind pressure pulses (as in the simulation) and interplanetary shock waves. All these events can play roles in driving intense space weather. We highlight this more clearly now in the introduction on lines 42-46.

There are, however, observational challenges, in general, in unambiguously showing that a surface eigenmode has been excited by such a driver, which is what we were describing in our previous response. The observational criteria established by Archer et al. (2019) involve simultaneous observations of the upstream driver, magnetopause boundary motion, and ULF waves. This is clearly a multi-spacecraft configuration which is not always available. However, the availability of correctly placed spacecraft does not preclude that the physical mechanism may still be regularly occurring.

Our other point was that the surface eigenmode occurs at the lowest frequencies present within the magnetosphere, typically below 2 mHz. Such low frequencies are often not focused on in the literature since in reality it may, in general, be difficult to distinguish potential signals from turbulence or noise at these frequencies. This can be understood through Figure 2, where the confidence intervals for autoregressive noise are largest at the lowest frequencies. By choosing an event where there are little turbulent pressure variations upstream following the impulse simply maximises the likelihood of observing the low frequency response. All these points have been

made in the Discussion (lines 354-368), though we now also add some of them to the introduction too (lines 70-72).

We therefore stress to the reviewer that this event is not in itself unique and it is likely that surface eigenmodes may occur regularly. Indeed, we noted that further instances of Poynting fluxes directed towards the subsolar point following jets were observed on this very day and we presented evidence of this through Figure 2 which covers a wider time interval. Detecting surface eigenmodes in general though remains observationally challenging due to the points mentioned. Future work could look to develop less restrictive observational criteria for the detection of MSE and/or undertake a statistical study of MSE occurrence in response to different driving events. We have added this on lines 368-371.

The significance and importance for the system of the results presented have been discussed in terms of the well-accepted paradigm of tailward propagation in global magnetospheric dynamics, as established in the introduction. Our results from observations, simulations, and theory all show propagation in the magnetosphere in opposition to and balancing the tailward flow of the magnetosheath outside the boundary. This is an unexpected result which does not confirm with current models of global magnetospheric dynamics. However, the result can be fairly simply understood through Figure 6, which reveals that the magnetopause may form an azimuthally stationary surface wave over a wide local time range on the dayside.

Such a response in near-Earth space will have wide-ranging impacts on the radiation belts and ionosphere unlike that expected from current models of magnetospheric dynamics. These have been discussed on lines 372-390. We have also raised the likely universality of surface eigenmodes at other planetary magnetospheres and potential applications to other environments that exhibit surface waves, such as coronal loops (lines 398-411).

The filtering applied doesn't allow one to understand the dynamics of the system - I think the presentation in the time-frequency domain can be more helpful here.

We have added a supplementary figure for THA and THE (THD proved similar to THE) showing the full dynamic spectra of all quantities considered using a continuous wavelet transform. This reveals the 1.8 MHz fundamental mode MSE (clearest in the compressional magnetic field components at both spacecraft) and 3.3 MHz second harmonic MSE (e.g. in the perpendicular components of the magnetic field), as well as a 6.7 MHz fundamental toroidal standing Alfvén wave at THA (azimuthal velocity / radial electric field), as further discussed in Archer et al. (2019). We also include Poynting fluxes computed from these wavelet transforms. These largely highlight the main results as presented through smoothing, i.e. that the Poynting fluxes are directed toward the magnetopause radially and towards the subsolar point azimuthally. However, the wavelet transforms offer some further clarity to the Poynting flux in the field-aligned direction, which we shall address after the subsequent reviewer comment.

I didn't understand the answer to my comment about the observed phase shift at THAA and THE - this could be clear if the authors present virtual spacecraft time-series from two points corresponding to the THEMIS spacecraft locations.

We apologise that we were not clear enough in our response. Our main point is that wave reflection at the ionospheres is not perfect nor is the absorption north-south symmetric. This will result in a superposition of standing and propagating components, thereby yielding some resultant Poynting fluxes along the magnetic field. This is well described in Allan (1982) applied to standing Alfvén waves, where they state that asymmetric conductances in the ionospheres (e.g.

due to a dipole tilt) will result in a “null point”, shifted from the near nodes/antinodes of the standing wave, from which wave energy propagates on either side to the ionospheres, where the energy dissipates. This is the reason why an eigenmode may still have some resultant Poynting flux along the field.

We include below Figure 4 of Allan (1982) which illustrates this effect for a second harmonic toroidal standing Alfvén wave in a dipole magnetic field. The top panel shows amplitudes of perturbations, revealing a node in the radial electric field and antinode in the azimuthal magnetic field near the equator, as expected for a standing wave. The bottom panel depicts the relative phases of these perturbations. This shows that the wave damping and asymmetric reflection by the northern and southern ionospheres results in more smoothly varying phases, rather than perfect quadrature at all points as would be expected for a perfectly standing wave. The consequence is that a “null point” occurs (vertical dashed line), slightly offset from the location of the node, either side of which there is a resultant Poynting flux directed to the ionospheres (arrows). The location of this null point will vary depending on the wave harmonic as well as both ionospheric boundary conditions on the field line in question. We have explained this in the revised manuscript now on lines 149-151.

Our other point was about the localised driver. This can intuitively be understood when considering standing waves excited by a pulse located slightly off the symmetry axis. There will be a slight mistiming between the surface wave which propagates towards and reflects off of the northern ionosphere compared to the southern one, occurring around one bounce period later, which can again lead to a net Poynting flux due to the phase differences. Of course, one can decompose this asymmetric initial perturbation on the boundary into its normal modes along the field to describe this mathematically. The decomposition of the observations into normal modes (namely the fundamental and second harmonic MSE), thanks to the wavelet transform the reviewer suggested, actually helps us better understand the differences between THE/THD and THA in the field-aligned Poynting fluxes. This is shown in Supplementary Figure 1c.

The direction of the time-averaged Poynting flux at the fundamental MSE frequency of 1.8 mHz has a component in the direction of the geomagnetic field at all spacecraft, resulting in similar directions between the Poynting vectors with only 26 degree differences (within expected experimental errors). This shows that the phases between spacecraft are consistent at this frequency and that they were located above this harmonic's "null point".

However, for the second harmonic at 3.3 mHz, while the Poynting vectors are similarly directed in the equatorial plane (to within 9 degrees), along the field we find that THE/THD observed southward fluxes whereas THA observed a slightly northward flux (though this was not statistically significant). A second harmonic wave has a node in displacement near the equator. Thus, at this frequency, the spacecraft near the equator are sensitive to which side of the "null point" they are located, as described by Allan (1982). We therefore conclude that THA was likely near the second harmonic MSE's null point whereas THE/THD were below it. This explains the phase shift between the spacecraft at this frequency.

A brief discussion of all of the above points has been added to clarify the manuscript on lines 153-173.

Unfortunately, field-aligned Poynting fluxes are not present in the simulation as it is symmetric both in terms of driver, dipole configuration, and ionospheric conductances. Therefore, virtual spacecraft time-series will not assist in further interpreting the field-aligned Poynting flux. The time-series shown in Figure 5a is representative of THE's location being at the same local time and close to the magnetic equator. Currently it is not possible to drive global MHD models with localised pressure pulses, however, future simulations could investigate MSE under asymmetric conditions such as with different dipole tilts and ionospheric conductances.

We finally reiterate that none of the conclusions in our paper are contingent on the field-aligned Poynting flux (which depends on the relative phases between perpendicular components of the magnetic and electric fields with one another) as the focus is the azimuthal propagation of surface waves, which depends only on the relative phases of the perpendicular electric field to the compressional magnetic field variations. However, we have further clarified the results on the field-aligned Poynting flux in the revision.

I appreciate the revision of the movie presentation but it is still hard to get the point - I suggest highlighting the area and time supporting the authors' conclusion. Probably, as the system is symmetrical, only half might be presented to provide better resolution. The zoom of the area of interest with highlighting the phenomenon discussed can be helpful here.

We have added an additional movie which zooms in on regions in the XY plane showing separately the stationary and travelling wave regions along the magnetopause. Hopefully this is now clearer.

Reviewer #3

The authors have addressed all my comments and have greatly improved the manuscript by putting the new results in the context of the global magnetospheric dynamics. I can therefore recommend it for publication in Nature Communications after a final, minor suggestion.

We thank the reviewer for their positive assessment of our revisions.

To further emphasize importance for global magnetospheric dynamics and for global plasma circulation related to surface wave growth, my only remaining suggestion is to add somewhere,

when discussing magnetospheric impacts, on lines 37 to 44 the following references to the first works demonstrating a physical mechanism for mass transport at the magnetopause produced by the non-linear KH waves (reconnection in KH vortices) which could grow from surface waves when sufficient free energy is present to overcome magnetic tension or plasma compressibility.

We have added this to lines 35-37.

References

W. Allan (1982) Phase variation of ULF pulsations along the geomagnetic field-line. Planetary and Space Science, 30, 339-346, [https://doi.org/10.1016/0032-0633\(82\)90039-3](https://doi.org/10.1016/0032-0633(82)90039-3).

Archer, M.O., Hietala, H., Hartinger, M.D. et al. Direct observations of a surface eigenmode of the dayside magnetopause. Nat Commun 10, 615 (2019). <https://doi.org/10.1038/s41467-018-08134-5>

Plaschke, F., Hietala, H., Angelopoulos, V., and Nakamura, R. (2016), Geoeffective jets impacting the magnetopause are very common, J. Geophys. Res. Space Physics, 121, 3240– 3253, doi:10.1002/2016JA022534.

REVIEWERS' COMMENTS

Reviewer #1 (Remarks to the Author):

The authors have replied to my comments and I recommend publication in present form.

Reviewer #2 (Remarks to the Author):

The authors have mostly addressed my comments. I think it can be recommended for publication in Nature Communications.

Reviewer #3 (Remarks to the Author):

Authors employ multi-spacecraft observations, global MHD simulations and analytic MHD theory to understand the somewhat puzzling observation of sunward Poynting-flux of the lowest frequency normal mode of the magnetopause surface waves.

The findings are novel and bring new insight into magnetospheric response to external driving. The results may be important also for other systems in our solar system or elsewhere.

The authors have performed two substantial revisions to address the questions by 3 reviewers. In my opinion authors have adequately addressed the questions raised by reviewers and I can therefore recommend it for publication in Nature Communications.

Response to reviewers

Second revision of 'Magnetopause ripples going against the flow form azimuthally stationary surface waves' Archer et al.

We would like to thank the reviewers for their time in providing comments on the various iterations of our manuscript. Please find our responses below to the final set of reviews.

Reviewer #1

The authors have replied to my comments and I recommend publication in present form.

We thank the reviewer for their assessment.

Reviewer #2

The authors have mostly addressed my comments. I think it can be recommended for publication in Nature Communications.

We thank the reviewer for their assessment.

Reviewer #3

Authors employ multi-spacecraft observations, global MHD simulations and analytic MHD theory to understand the somewhat puzzling observation of sunward Poynting-flux of the lowest frequency normal mode of the magnetopause surface waves.

The findings are novel and bring new insight into magnetospheric response to external driving. The results may be important also for other systems in our solar system or elsewhere.

The authors have performed two substantial revisions to address the questions by 3 reviewers. In my opinion authors have adequately addressed the questions raised by reviewers and I can therefore recommend it for publication in Nature Communications.

We thank the reviewer for their assessment.